# `CoUn`: Empowering Machine Unlearning via Contrastive Learning

**Yasser H. Khalil**     **Mehdi Setayesh**     **Hongliang Li**

Huawei Noah's Ark Lab, Montreal, Canada
{yasser.khalil1, mehdi.setayesh1, hongliang.li2}@huawei.com

## Abstract

Machine unlearning (MU) aims to remove the influence of specific "forget" data from a trained model while preserving its knowledge of the remaining "retain" data. Existing MU methods based on label manipulation or model weight perturbations often achieve limited unlearning effectiveness. To address this, we introduce `CoUn`, a novel MU framework inspired by the observation that a model retrained from scratch using only retain data classifies forget data based on their semantic similarity to the retain data. `CoUn` emulates this behavior by adjusting learned data representations through contrastive learning (CL) and supervised learning, applied exclusively to retain data. Specifically, `CoUn` (1) leverages semantic similarity between data samples to indirectly adjust forget representations using CL, and (2) maintains retain representations within their respective clusters through supervised learning. Extensive experiments across various datasets and model architectures show that `CoUn` consistently outperforms state-of-the-art MU baselines in unlearning effectiveness. Additionally, integrating our CL module into existing baselines empowers their unlearning effectiveness.

## 1   Introduction

The widespread adoption of machine learning (ML) has raised concerns regarding data privacy and regulatory compliance, such as the General Data Protection Regulation (GDPR) [1, 2]. Machine unlearning (MU) [3, 4, 5] addresses these concerns by removing the influence of specific training data (i.e., *forget data*) from a trained model, termed the **Original model**, while preserving the knowledge of the remaining data (i.e., *retain data*). Retraining the model from scratch on retain data is considered *exact unlearning* [6, 7], termed the gold-standard **Retrain model**. While exact unlearning is effective, it is computationally inefficient. Alternatively, *approximate unlearning* aims to efficiently achieve an unlearned model that performs approximately the same as the Retrain model [8, 9, 10].

To develop an effective approximate unlearning algorithm, we first analyze how the Retrain model classifies forget data. Figure 1 illustrates the representation space of two Retrain models: one trained without 'truck' class samples (i.e., *class-wise forgetting*) and another trained without 10% randomly selected samples (i.e., *random forgetting*). Our analysis reveals that *the Retrain model classifies forget samples into clusters of retain samples that exhibit the highest semantic similarity to them.* For instance, in class-wise forgetting, forget 'truck' samples are mostly misclassified as semantically similar clusters, like 'automobile' (69.32%), 'airplane' (13.47%), and 'ship' (12.60%), with no correct classifications due to the absence of a 'truck' cluster. In random forgetting, 97.42% of forget 'truck' samples are correctly classified as 'truck' due to their semantic similarity with 'truck' samples in the retain data, while most of the others are misclassified as 'automobile' (1.23%), 'airplane' (0.38%), and 'ship' (0.4%).

39th Conference on Neural Information Processing Systems (NeurIPS 2025).

Thus, to closely match the performance of exact unlearning, approximate unlearning should adjust the learned data representations of the Original model such that: ❶ retain samples are correctly classified, and ❷ forget samples are pushed into clusters of other retain samples that exhibit the highest semantic similarity to them. In random forgetting, these clusters may be the same as or different from the original clusters of the forget samples; however, in class-wise forgetting, they must be different from the original clusters of the forget samples.

Existing MU methods [8, 9, 10, 11, 12] attempt to close the performance gap between approximate and exact unlearning, focusing on three

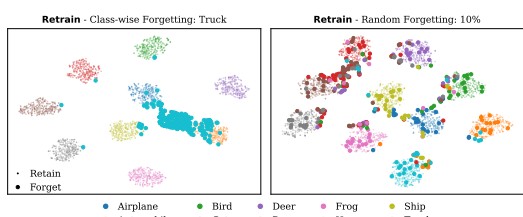

Figure 1: **Representation space of the Retrain model** trained with ResNet-18 and CIFAR-10, excluding 'truck' class samples (*left*) and excluding 10% randomly selected samples (*right*). Small dots represent retain samples from different clusters, while larger dots indicate forget samples classified into clusters of retain samples that exhibit the highest semantic similarity to them.

criteria: ❶ **good forget quality** (evaluating misclassification of forget data and success in membership inference attacks [9, 11, 13]), ❷ **high utility** (maintaining high accuracy on retain data and generalizing well to test data), and ❸ **efficiency**. Prior methods [14, 9, 15, 16] often rely on label manipulation, assigning forget samples to semantically inconsistent or random clusters to increase misclassification. However, this strategy can degrade the performance on retain data, thereby reducing utility. Moreover, these methods require access to forget data, which may not always be available—particularly in federated unlearning settings where clients may leave the network after requesting unlearning [17, 8]. Recent methods focus on model weight perturbations [8, 10], and eliminate the need for forget data. Yet, excessive perturbations can compromise utility by erasing important knowledge, while weak perturbations degrade forget quality. Thus, inadequate perturbations limit unlearning effectiveness.

In light of the above, we propose `CoUn` (*Contrastive learning for empowering Unlearning*), a novel MU framework that leverages contrastive learning (CL) and supervised learning, applied exclusively to retain data. `CoUn` exploits the fact that the learned representations of the Original model has already captured the semantic similarities among samples. Consequently, adjusting the retain representations during unlearning indirectly influences the forget representations. In `CoUn`, the CL module (Figure 2) pulls together representations of augmented views of the same retain samples (positives) while pushing apart representations of different retain samples (negatives) [18, 19, 20, 21]. However, false negatives, which are samples that belong to the same cluster as positives but are treated as negatives, can cause clusters to overlap. This phenomenon is known as *cluster collision* [21, 22, 23]. As a result, forget representations are indirectly pushed toward clusters of other retain samples that exhibit the highest semantic similarity to them, thereby improving forget quality. Additionally, `CoUn` preserves high utility by mitigating cluster collision among retain representations through supervised learning. This adjustment of forget representations toward semantically similar retain representations, while maintaining the separation of retain clusters, enables effective unlearning. Our key **contributions** are:

• We introduce `CoUn`, a novel MU framework that achieves effective unlearning by adjusting learned data representations based on semantic similarity using CL and supervised learning on retain data.

• We provide empirical insights into how `CoUn` effectively unlearns by pushing forget representations into clusters of semantically similar retain representations. We also present a theoretical analysis showing that `CoUn` induces a higher misclassification rate on forget data while maintaining low misclassification rate on retain data, thereby contributing to better forget quality and utility.

• We validate `CoUn` across different datasets, model architectures, and forgetting scenarios, showing a reduced performance gap with exact unlearning compared to MU baselines. We also demonstrate that integrating our CL module into existing baselines empowers their unlearning effectiveness.

## 2   Related Work

**Label Manipulation.**   Existing MU methods attempt to unlearn by manipulating labels of forget data, thereby misleading the model into learning incorrect labels. **NegGrad** [24, 25] achieves this by applying gradient ascent on the forget data. **NegGrad+** [14] combines fine-tuning (minimizing loss with respect to retain data) and gradient ascent (maximizing loss with respect to forget data). Both NegGrad and NegGrad+ exemplify label manipulation strategies by modifying the model's

output distribution of the forget data through targeted gradient-based adjustments. **Amnesiac** [15] and **SalUn** [9] randomly relabel the forget data and then apply joint optimization on both retain and forget data. **BadT** [16] uses knowledge distillation from two teacher models (Original and Random models) into the unlearned model, while **SCRUB** [14] extends BadT by incorporating NegGrad+. Both BadT and SCRUB manipulate the model's output distributions of forget data by using a random teacher, effectively altering the perceived labels of the forget data. However, such methods that assign forget samples to semantically inconsistent or random classes to increase misclassification rates often degrade performance on retain data, resulting in lower utility. Moreover, as shown in Figure 1, exact unlearning does not aim to misclassify all forget samples, as some can be correctly classified.

**Model Weight Perturbation.**   Recent methods perturb the model's weights to achieve unlearning. $\ell_1$**-sparse** [10] introduces an $\ell_1$ regularization term in the objective function, inspired by model pruning [26, 27]. In addition to random labeling, SalUn [9] adjusts model weight parameters using gradient-based saliency masks. **SSD** [28] uses the Fisher information matrix to identify and dampen weight parameters critical to forget data. **NoT** [8] negates layer-wise weights to support unlearning. However, inadequate perturbations reduce unlearning effectiveness. Excessive perturbations degrade utility by erasing essential knowledge, while insufficient perturbations fail to properly eliminate the influence of forget data, thereby compromising forget quality.

**Contrastive Learning.**   The goal of CL is to maximize similarity between augmented views of the same sample while minimizing similarity between different samples [18, 19, 20, 29, 30, 31, 32]. CL's ability to adjust representations aligns with the objective of MU, which seeks to disrupt the representations of forget data. This work focuses on CL with the InfoNCE loss, as used in SimCLR [20]. One method that applies CL for unlearning pushes forget samples away from retain samples of the same class and pulls them closer to retain samples of different classes [33]. However, this method harms the model's utility by pushing forget samples away from their original clusters, resulting in high misclassification rates. `CoUn` differs from [33] by utilizing semantic similarity and cluster collision for effective unlearning, instead of forcing forget samples to be misclassified. Similar to exact unlearning, `CoUn` yields a higher misclassification rate for forget data while maintaining low misclassification rate on retain data. Further discussion is provided in Appendix A.

## 3   Methodology

We begin by formulating the MU problem in Section 3.1. Section 3.2 then introduces our proposed MU framework, `CoUn`. Section 3.3 follows with empirical insights demonstrating how CL and supervised learning adjusts the representation space to facilitate effective unlearning. Finally, Section 3.4 provides theoretical insights.

### 3.1   Machine Unlearning: Background

Given a dataset $\mathcal{D}$, partitioned into **forget data** $\mathcal{D}_u$ and **retain data** $\mathcal{D}_r = \mathcal{D} \setminus \mathcal{D}_u$, the goal of MU is to transform an **Original model** $\boldsymbol{\theta}_o$, trained on $\mathcal{D}$, into an **unlearned model** $\boldsymbol{\theta}_u$ that effectively removes the influence of $\mathcal{D}_u$. We define the forget data ratio as: $|\mathcal{D}_u|/|\mathcal{D}| \times 100$.

In *exact unlearning*, $\boldsymbol{\theta}_u$ is obtained by training a randomly initialized model only on $\mathcal{D}_r$, yielding the gold-standard **Retrain model**. While this ensures precise unlearning, it incurs significant computational cost. In contrast, *approximate unlearning* methods aim to obtain $\boldsymbol{\theta}_u$ more efficiently, often at the expense of reduced effectiveness, resulting in a performance gap relative to the Retrain model. In approximate unlearning, $\boldsymbol{\theta}_u$ is initialized with the parameters of $\boldsymbol{\theta}_o$.

### 3.2   `CoUn` **Overview**

`CoUn` is a CL-based MU framework designed to adjust the learned representation space of both retain and forget data for effective unlearning. The framework of `CoUn` incorporates two key components: ❶ **contrastive learning** and ❷ **supervised learning**. The overall architecture of `CoUn` is illustrated in Figure 2. In `CoUn`, CL is applied on retain data via a *CL module* to adjust their representations by pulling the representations of two augmented views of the same sample (positives) closer together, while pushing representations of different samples (negatives) further apart [18, 19, 20, 21].

Let $(\boldsymbol{I}, \boldsymbol{Y})$ denote a batch of images and their corresponding labels sampled from $\mathcal{D}_r$. We have $(\boldsymbol{I}, \boldsymbol{Y}) = \{(\boldsymbol{i}_n, \boldsymbol{y}_n)\}_{n=1}^N$, where $N$ is the batch size and $(\boldsymbol{i}_n, \boldsymbol{y}_n) \in \mathcal{D}_r$. We consider that $\boldsymbol{Y} \in \mathbb{R}^{N \times K}$ represents a batch of one-hot encoded target labels, with $K$ denoting the number of classes. Two augmented views, $\boldsymbol{X} = t(\boldsymbol{I})$ and $\boldsymbol{X}' = t'(\boldsymbol{I})$ are generated using transformations $t, t' \sim \mathcal{T}$, respectively, where $\mathcal{T}$ is a transformation distribution combining multiple image augmentations (e.g. random cropping, random horizontal flipping, and color normalization). These augmented

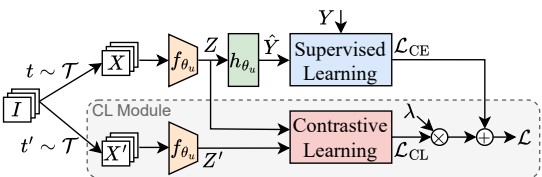

Figure 2: CoUn **framework.** Two augmented views are generated from a batch of retain image samples $\boldsymbol{I}$. These views are processed by the feature extractor $f_{\boldsymbol{\theta}_u}$, yielding retain representations $(\boldsymbol{Z}, \boldsymbol{Z}')$. A *CL module* adjusts the representations, while supervised learning applied via the classifier head $h_{\boldsymbol{\theta}_u}$ enforces their cluster separation.

views are encoded by the feature extractor $f_{\boldsymbol{\theta}_u}$, producing **representations** $\boldsymbol{Z} = f_{\boldsymbol{\theta}_u}(\boldsymbol{X})$ and $\boldsymbol{Z}' = f_{\boldsymbol{\theta}_u}(\boldsymbol{X}') \in \mathbb{R}^{N \times D}$, where $D$ is the representation dimension. The representations are compared using cosine similarity, a commonly employed measure in CL to assess the similarity between normalized vectors. The cosine similarity between two normalized vectors $\boldsymbol{w}$ and $\boldsymbol{v}$ is computed as $\boldsymbol{w} \cdot \boldsymbol{v} = \boldsymbol{w}^T \boldsymbol{v}$. Let $\boldsymbol{z}_n$ and $\boldsymbol{z}'_n$ denote the $n$-th row of the representation matrices $\boldsymbol{Z}$ and $\boldsymbol{Z}'$, respectively. For a given representation $\boldsymbol{z}_n$ corresponding to image $\boldsymbol{i}_n$, the CL loss for the positive pair of representations $(\boldsymbol{z}_n, \boldsymbol{z}'_n)$ is calculated as:

$$l(\boldsymbol{z}_n) = -\log \frac{\exp(\boldsymbol{z}_n \cdot \boldsymbol{z}'_n / \tau)}{\sum_{j=1}^N \exp(\boldsymbol{z}_n \cdot \boldsymbol{z}'_j / \tau)}, \tag{1}$$

where $\tau$ is a temperature constant. Each representation $\boldsymbol{z}'_j \neq \boldsymbol{z}'_n$ in the batch is considered a negative for $\boldsymbol{z}_n$. This CL loss encourages positive pairs of representations to be closer in the representation space while pushing them away from negative representations. Following SimCLR [20], we employ a symmetric formulation for the overall CL loss, defined as:

$$\mathcal{L}_{\mathrm{CL}}(\boldsymbol{Z}, \boldsymbol{Z}') = \frac{1}{N} \sum_{n=1}^N \big(l(\boldsymbol{z}_n) + l(\boldsymbol{z}'_n)\big). \tag{2}$$

Since CoUn samples a random batch, it does not explicitly select negatives. Consequently, false negatives, which are negative image samples sharing the same class label as the positive image sample, may be present. Specifically, for each image sample $\boldsymbol{i}_n$, images $\boldsymbol{i}_j$ in the batch (with $j \neq n$) are false negatives if $\boldsymbol{y}_j = \boldsymbol{y}_n$. Including false negatives in CL introduces the *cluster collision* problem [21, 22, 23], where samples from the same cluster are pushed apart, potentially bringing them closer to semantically similar samples from other clusters, thus causing cluster overlap. However, false negatives that are more semantically similar with other samples within their own clusters will remain in their original clusters. In other words, not all false negatives will leave their original clusters.

Moreover, since the Original model is trained on both retain and forget data, which share semantic information, this information is captured in the model's weights and reflected in its learned representations. Arora et al. [34] provide theoretical and empirical evidence showing that learned representations connect similarity in the training data to the semantic information that is implicitly present in downstream tasks. Their work also establishes provable guarantees on the performance of such representations in downstream settings. This supports the observation that the Retrain model—trained solely on retain data—classifies forget samples based on semantic similarity, and this explains why in random forgetting it can correctly classify some of the forget data. From the Retrain model's perspective, retain data serve as the training set, and forget data act as the downstream task. Consequently, any adjustment to retain representations will indirectly influence forget representations.

Building on this principle, when the learned retain representations are adjusted using CL, forget representations are indirectly pushed toward clusters of other retain samples that exhibit the highest semantic similarity to them. In random forgetting scenario, these retain samples may belong to clusters that are either the same as or different from the original clusters of the forget samples. In contrast, in class-wise forgetting scenario, they necessarily belong to different clusters. As a result, the CL module in CoUn can induce cluster collisions among both retain and forget representations. While this improves forget quality, it also degrades model's utility. To address this, CoUn mitigates the impact of cluster collision on retain representations by applying supervised learning to the retain

data. This ensures that retain representations are preserved within their respective clusters, thereby maintaining high model utility.

Let $\hat{\boldsymbol{Y}} \in \mathbb{R}^{N \times K}$ represent the model predictions. We have $\hat{\boldsymbol{Y}} = h_{\boldsymbol{\theta}_u}(\boldsymbol{Z})$, where the classifier head $h_{\boldsymbol{\theta}_u}$ takes representations $\boldsymbol{Z}$ as input to produce predictions $\hat{\boldsymbol{Y}}$. Cross-entropy (CE) loss is used in `CoUn` for supervised learning to maintain cluster separation for retain samples, and is defined as:

$$\mathcal{L}_{\text{CE}}(\boldsymbol{Y}, \hat{\boldsymbol{Y}}) = -\frac{1}{N} \sum_{n=1}^{N} \sum_{k=1}^{K} \boldsymbol{y}_{n,k} \log(\hat{\boldsymbol{y}}_{n,k}). \tag{3}$$

Therefore, the final loss function combines CE and CL losses, weighted by a scaling factor $\lambda$:

$$\mathcal{L} = \mathcal{L}_{\text{CE}} + \lambda \mathcal{L}_{\text{CL}}. \tag{4}$$

Appendix B presents the pseudo-code and the PyTorch implementation of `CoUn`.

### 3.3 Empirical Analysis

Figure 3 visualizes the representation space of four $\boldsymbol{\theta}_u$ models produced by the fine-tune (FT) and `CoUn` unlearning methods under class-wise and random forgetting scenarios. All four $\boldsymbol{\theta}_u$ models are initialized from $\boldsymbol{\theta}_o$, which is trained on the entire CIFAR-10 training data using ResNet-18. FT reduces the influence of $\mathcal{D}_u$ through catastrophic forgetting by fine-tuning $\boldsymbol{\theta}_o$ on $\mathcal{D}_r$. Although the CL module in `CoUn` is applied only to retain data, the visualization shows that the forget representations are affected—they are pushed toward clusters of other retain samples that exhibit the highest semantic similarity to them. Additionally, we can see that for random forgetting these clusters may either be the same as or different from the original clusters of the forget samples; and for class-wise forgetting, they are necessarily different. Meanwhile, retain representations remain well-clustered due to supervised learning. Notably, under class-wise forgetting setting, FT performs comparably to Retrain (Figure 1), due to the relatively low entanglement between $\mathcal{D}_r$ and $\mathcal{D}_u$ in this setting, making it a simpler unlearning task than random forgetting [11].

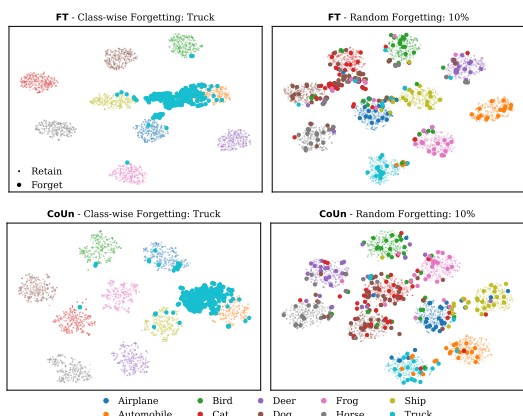

Figure 3: **Representation space of FT and `CoUn` unlearned models** (*rows*). *Columns* correspond to two forgetting scenarios: class-wise ('truck') and random (10% forget ratio). The Original model is trained on CIFAR-10 using ResNet-18. Small dots represent retain samples from different clusters, while larger dots indicate forget samples classified into the corresponding clusters. To achieve effective unlearning, `CoUn` adjusts representations based on semantic similarity, pushing forget representations into clusters of other retain samples with the highest semantic similarity to them, and preserving retain representations within their clusters.

Table 1 further confirms that `CoUn` more effectively classifies forget samples based on semantic similarity than FT, achieving performance closer to that of the Retrain model in both class-wise and random forgetting scenarios. In particular, `CoUn` produces forget sample predictions that more closely align with those of the Retrain model. Further experiments is provided in Appendix C.

Table 1: **Predictions of forget 'truck' samples based on most semantically similar classes**. Experiments are conducted using CIFAR-10 and ResNet-18. The difference ($\Delta$) and the (best) average difference between each method and Retrain are reported.

| Forgetting Scenario | Method | Predictions (%) - ($\Delta \downarrow$) | | | | Avg. Diff. $\downarrow$ |
| --- | --- | --- | --- | --- | --- | --- |
| | | Truck | Automobile | Airplane | Ship | |
| Class ('truck') | Retrain | 0.00 (0.00) | 69.32 (0.00) | 13.47 (0.00) | 12.60 (0.00) | 0.00 |
| | FT | 0.00 (0.00) | 70.29 (0.97) | 12.38 (1.09) | 13.12 (0.52) | 0.65 |
| | CoUn | 0.00 (0.00) | 69.60 (0.28) | 13.96 (0.49) | 13.13 (0.53) | 0.33 |
| Random (10%) | Retrain | 97.42 (0.00) | 1.23 (0.00) | 0.38 (0.00) | 0.40 (0.00) | 0.00 |
| | FT | 98.12 (0.70) | 0.75 (0.48) | 0.36 (0.02) | 0.32 (0.08) | 0.32 |
| | CoUn | 97.84 (0.42) | 0.99 (0.24) | 0.32 (0.06) | 0.42 (0.02) | 0.19 |

### 3.4 Theoretical Analysis

In this section, we provide a theoretical analysis showing that `CoUn` yields a higher misclassification rate on forget data compared to retain data. Achieving a higher misclassification rate for forget data while maintaining a low misclassification rate for retain data contributes to both good forget quality and high model utility. Recall that $K$ denotes the number of classes in the dataset. In particular, each

data sample belongs to one of classes $C_1, C_2, \ldots, C_K$. For a given transformation distribution $\mathcal{T}$, let $d_{\mathcal{T}}(\boldsymbol{i}_1, \boldsymbol{i}_2) = \min_{\boldsymbol{x}_1 \in t(\boldsymbol{i}_1), \boldsymbol{x}_2 \in t'(\boldsymbol{i}_2)} \|\boldsymbol{x}_1 - \boldsymbol{x}_2\|$ denote the augmented distance between two images $\boldsymbol{i}_1$ and $\boldsymbol{i}_2$, where $t, t' \sim \mathcal{T}$. We consider that a $(\sigma, \delta)$-augmentation is being used. That is, for each class $C_k$, there exists a subset $C_k^0 \subseteq C_k$, such that both $P[\boldsymbol{i} \in C_k^0] \geq \sigma P[\boldsymbol{i} \in C_k]$ and $\sup_{\boldsymbol{i}_1, \boldsymbol{i}_2 \in C_k^0} d_{\mathcal{T}}(\boldsymbol{i}_1, \boldsymbol{i}_2) \leq \delta$ hold, where $\sigma \in (0, 1]$. Let $\boldsymbol{\mu}_k$ denote the center of class $C_k$. We have $\boldsymbol{\mu}_k = \mathbb{E}_{\boldsymbol{i} \in C_k} \mathbb{E}_{\boldsymbol{x} \in t(\boldsymbol{i})}[f_{\boldsymbol{\theta}_u}(\boldsymbol{x})]$. Let $R[\epsilon]$ denote the probability that, when a sample is drawn from the dataset, the distance between the encoder representations of its two augmented views exceeds $\epsilon$, with a small value of $R[\epsilon]$ indicating good alignment in the representation space. We have

$$R[\epsilon] = P \left[ \boldsymbol{i} \in \bigcup_{k=1}^{K} C_k \,\middle|\, \sup_{\substack{\boldsymbol{x} = t(\boldsymbol{i}), \boldsymbol{x}' \in t'(\boldsymbol{i}) \\ t, t' \sim \mathcal{T}}} \|f_{\boldsymbol{\theta}_u}(\boldsymbol{x}) - f_{\boldsymbol{\theta}_u}(\boldsymbol{x}')\| > \epsilon \right]. \tag{5}$$

The following theorem shows the generalization ability of CL by providing an upper bound for the misclassification rate:

**Theorem 1.** *For an L-Lipschitz feature extractor $f_{\boldsymbol{\theta}_u}$, given a $(\sigma, \delta)$-augmentation used in CL, if*

$$\boldsymbol{\mu}_l^T \boldsymbol{\mu}_k < \frac{1}{2} \min_{k'} \|\boldsymbol{\mu}_{k'}\|^2 - \rho^{max}(\sigma, \delta, \epsilon) - \sqrt{2\rho^{max}(\sigma, \delta, \epsilon)} \tag{6}$$

*holds for any pair of $(l, k)$ with $l \neq k$, then the misclassification rate of classifier head $h_{\boldsymbol{\theta}_u}$ is $Err(h_{\boldsymbol{\theta}_u}) \leq (1-\sigma) + R[\epsilon]$, where $\rho^{max}(\sigma, \delta, \epsilon) = 2(1-\sigma) + \frac{R[\epsilon]}{\min_{k'} P[\boldsymbol{i} \in C_{k'}]} + \sigma(L\delta + 2\epsilon)$. Note that better alignment (i.e., smaller $R[\epsilon]$) and sharper concentration of augmented samples (i.e., larger $\sigma$ for a given $\delta$) result in a lower value of $\rho^{max}(\sigma, \delta, \epsilon)$.*

*Proof.* Refer to Theorem 1 in [35]. □

In an unlearning problem, the training data is divided into retain and forget data. In `CoUn`, the feature extractor $f_{\boldsymbol{\theta}_u}$ and the classifier head $h_{\boldsymbol{\theta}_u}$ are trained solely based on the samples in the retain data. From Theorem 1, it can be inferred that $\sigma, \delta, \epsilon, L$ are not dependent on the samples in the retain and forget data. Furthermore, in a random forgetting scenario, $\boldsymbol{\mu}_k, k \in \{1, \ldots, K\}$ would be relatively similar for the forget and retain data. Thus, the only parameter that may have different values for retain and forget data is $R[\epsilon]$. Let $R_r[\epsilon]$ and $R_u[\epsilon]$ characterize $R[\epsilon]$ for retain and forget data, respectively. We have the following lemma:

**Lemma 1.** *Considering a feature extractor $f_{\boldsymbol{\theta}_u}$ which is trained by both CL and supervised learning only on the retain data, we have $R_r[\epsilon] < R_u[\epsilon]$.*

*Proof.* See Appendix D. □

From Theorem 1, we can see that the misclassification rate of classifier head $h_{\boldsymbol{\theta}_u}$ in `CoUn` is obtained as follows: $Err(h_{\boldsymbol{\theta}_u}) \leq (1-\sigma) + R_r[\epsilon]$. From Lemma 1, we can see that $R_r[\epsilon] < R_u[\epsilon]$. Thus, the upper-bound on the misclassification rate of classifier $h_{\boldsymbol{\theta}_u}$ cannot be met for the forget data. This implies that `CoUn` provides a higher misclassification rate on the forget data compared to the retain data. The misclassification rate on retain data remains low because supervised learning is applied to $f_{\boldsymbol{\theta}_u}$ using only retain data. Our experiments also confirm this.

## 4 Experiments

### 4.1 Experimental Setup

**Datasets and Model Architectures.** We evaluate `CoUn` on three datasets: CIFAR-10/100 [36] and TinyImageNet [37], using three model architectures: ResNet-18 [38], VGG-16 [39], and ViT [40]. Additional details regarding the model architectures are provided in Appendix E.1.

**Baselines.** We compare `CoUn` with the following baselines: ❶ **Retrain**: Training from scratch on retain data $\mathcal{D}_r$; ❷ **FT**: Fine-tuning the Original model $\boldsymbol{\theta}_o$ on $\mathcal{D}_r$; ❸ **NegGrad+** (NeurIPS, 2023); ❹ $\ell_1$-**sparse** (NeurIPS, 2023); ❺ **SalUn** (ICLR, 2024); and ❻ **NoT** (CVPR, 2025).

Table 2: **Performance comparison of** `CoUn` **to the baseline methods** with **10% random data removal**. The gap ($\Delta$) and the (best) average gap between each method and the Retrain model are reported.

| Dataset & Model | Method | Accuracy (%) | | | Efficacy (%) | Avg. Gap ↓ | Comp. Cost (PFLOPs) ↓ |
|---|---|---|---|---|---|---|---|
| | | Retain ($\Delta\downarrow$) | Unlearn ($\Delta\downarrow$) | Test ($\Delta\downarrow$) | MIA ($\Delta\downarrow$) | | |
| CIFAR-10 ResNet-18 | Retrain | $100.00_{\pm 0.00}(0.00)$ | $4.81_{\pm 0.27}(0.00)$ | $94.67_{\pm 0.24}(0.00)$ | $11.02_{\pm 0.58}(0.00)$ | 0.00 | 27.37 |
| | FT | $99.99_{\pm 0.00}(0.01)$ | $3.76_{\pm 0.31}(1.05)$ | $94.70_{\pm 0.14}(0.03)$ | $9.51_{\pm 0.28}(1.51)$ | 0.65 | 6.32 |
| | NegGrad+ | $99.95_{\pm 0.02}(0.05)$ | $4.82_{\pm 0.24}(0.01)$ | $94.32_{\pm 0.23}(0.35)$ | $9.09_{\pm 0.30}(1.93)$ | 0.58 | 6.02 |
| | $\ell_1$-sparse | $99.97_{\pm 0.01}(0.03)$ | $5.40_{\pm 0.40}(0.59)$ | $93.81_{\pm 0.21}(0.86)$ | $10.97_{\pm 0.35}(0.05)$ | 0.38 | 6.92 |
| | SalUn | $99.10_{\pm 0.35}(0.90)$ | $4.31_{\pm 0.42}(0.50)$ | $93.84_{\pm 0.27}(0.83)$ | $11.15_{\pm 2.04}(0.13)$ | 0.59 | 8.66 |
| | NoT | $99.99_{\pm 0.00}(0.01)$ | $4.19_{\pm 0.25}(0.62)$ | $94.65_{\pm 0.24}(0.02)$ | $10.45_{\pm 0.51}(0.57)$ | 0.30 | 7.52 |
| | CoUn | $99.99_{\pm 0.00}(0.01)$ | $4.12_{\pm 0.31}(0.69)$ | $94.57_{\pm 0.24}(0.10)$ | $10.81_{\pm 0.31}(0.21)$ | 0.25 | 8.02 |
| CIFAR-100 ResNet-18 | Retrain | $99.98_{\pm 0.00}(0.00)$ | $24.26_{\pm 0.53}(0.00)$ | $75.56_{\pm 0.26}(0.00)$ | $48.44_{\pm 0.36}(0.00)$ | 0.00 | 27.37 |
| | FT | $99.97_{\pm 0.00}(0.01)$ | $16.39_{\pm 0.60}(7.87)$ | $76.75_{\pm 0.25}(1.19)$ | $44.06_{\pm 0.58}(4.38)$ | 3.36 | 7.22 |
| | NegGrad+ | $99.96_{\pm 0.01}(0.02)$ | $30.09_{\pm 0.41}(5.83)$ | $75.46_{\pm 0.36}(0.10)$ | $47.72_{\pm 0.32}(0.72)$ | 1.67 | 7.62 |
| | $\ell_1$-sparse | $99.95_{\pm 0.01}(0.03)$ | $23.94_{\pm 0.50}(0.32)$ | $74.95_{\pm 0.32}(0.61)$ | $42.81_{\pm 0.56}(5.63)$ | 1.65 | 7.22 |
| | SalUn | $98.55_{\pm 0.18}(1.43)$ | $20.35_{\pm 1.31}(3.91)$ | $72.02_{\pm 0.45}(3.54)$ | $52.37_{\pm 1.82}(2.93)$ | 2.95 | 5.69 |
| | NoT | $99.97_{\pm 0.01}(0.01)$ | $17.99_{\pm 0.40}(6.27)$ | $76.27_{\pm 0.24}(0.71)$ | $44.28_{\pm 0.57}(4.16)$ | 2.79 | 7.22 |
| | CoUn | $99.97_{\pm 0.00}(0.01)$ | $22.01_{\pm 0.44}(2.25)$ | $72.88_{\pm 0.39}(2.68)$ | $47.82_{\pm 0.96}(0.62)$ | 1.39 | 9.63 |
| TinyImageNet ResNet-18 | Retrain | $99.98_{\pm 0.00}(0.00)$ | $36.16_{\pm 0.35}(0.00)$ | $63.82_{\pm 0.20}(0.00)$ | $63.73_{\pm 0.42}(0.00)$ | 0.00 | 218.98 |
| | FT | $99.98_{\pm 0.00}(0.00)$ | $32.76_{\pm 0.42}(3.40)$ | $64.65_{\pm 0.29}(0.83)$ | $56.93_{\pm 0.59}(6.80)$ | 2.76 | 60.16 |
| | NegGrad+ | $99.98_{\pm 0.00}(0.00)$ | $38.01_{\pm 0.32}(1.85)$ | $64.68_{\pm 0.26}(0.86)$ | $57.84_{\pm 0.47}(5.89)$ | 2.15 | 80.21 |
| | $\ell_1$-sparse | $99.96_{\pm 0.00}(0.02)$ | $36.96_{\pm 0.37}(0.80)$ | $62.62_{\pm 0.39}(1.20)$ | $56.74_{\pm 0.46}(6.99)$ | 2.25 | 60.16 |
| | SalUn | $98.52_{\pm 0.32}(1.46)$ | $34.03_{\pm 1.06}(2.13)$ | $61.21_{\pm 0.57}(2.61)$ | $67.72_{\pm 1.21}(3.99)$ | 2.55 | 51.08 |
| | NoT | $99.98_{\pm 0.00}(0.00)$ | $35.64_{\pm 0.71}(0.52)$ | $63.66_{\pm 0.70}(0.16)$ | $56.08_{\pm 0.93}(7.65)$ | 2.08 | 80.21 |
| | CoUn | $99.95_{\pm 0.01}(0.03)$ | $35.10_{\pm 0.30}(1.06)$ | $63.27_{\pm 0.12}(0.55)$ | $57.57_{\pm 0.17}(6.16)$ | 1.95 | 80.21 |
| CIFAR-100 VGG-16 | Retrain | $99.75_{\pm 0.07}(0.00)$ | $33.23_{\pm 0.38}(0.00)$ | $67.07_{\pm 0.57}(0.00)$ | $40.69_{\pm 0.40}(0.00)$ | 0.00 | 15.58 |
| | FT | $99.26_{\pm 0.05}(0.49)$ | $26.02_{\pm 0.55}(7.21)$ | $68.42_{\pm 0.32}(1.35)$ | $35.51_{\pm 0.62}(5.18)$ | 3.56 | 3.42 |
| | NegGrad+ | $94.92_{\pm 0.41}(4.83)$ | $35.44_{\pm 0.62}(2.21)$ | $65.54_{\pm 0.39}(1.53)$ | $40.67_{\pm 0.60}(0.02)$ | 2.15 | 3.42 |
| | $\ell_1$-sparse | $99.27_{\pm 0.04}(0.48)$ | $26.96_{\pm 0.66}(6.27)$ | $68.01_{\pm 0.37}(0.94)$ | $35.31_{\pm 0.50}(5.38)$ | 3.27 | 3.42 |
| | SalUn | $92.65_{\pm 0.47}(7.10)$ | $33.00_{\pm 0.88}(0.23)$ | $64.04_{\pm 0.33}(3.03)$ | $42.85_{\pm 1.48}(2.16)$ | 3.13 | 2.79 |
| | NoT | $96.17_{\pm 4.28}(3.58)$ | $30.11_{\pm 3.02}(3.12)$ | $66.75_{\pm 1.73}(0.32)$ | $36.47_{\pm 1.18}(4.22)$ | 2.81 | 4.28 |
| | CoUn | $99.82_{\pm 0.01}(0.07)$ | $32.37_{\pm 0.46}(0.86)$ | $63.80_{\pm 0.35}(3.27)$ | $39.64_{\pm 0.25}(1.05)$ | 1.31 | 5.71 |
| CIFAR-100 ViT | Retrain | $99.97_{\pm 0.00}(0.00)$ | $38.73_{\pm 0.69}(0.00)$ | $61.89_{\pm 0.62}(0.00)$ | $61.75_{\pm 0.33}(0.00)$ | 0.00 | 86.83 |
| | FT | $99.78_{\pm 0.04}(0.19)$ | $10.83_{\pm 0.41}(27.90)$ | $61.12_{\pm 0.45}(0.77)$ | $31.50_{\pm 0.42}(30.25)$ | 14.78 | 5.79 |
| | NegGrad+ | $99.88_{\pm 0.03}(0.09)$ | $45.26_{\pm 0.41}(6.53)$ | $59.33_{\pm 0.64}(2.56)$ | $55.00_{\pm 0.40}(6.75)$ | 3.98 | 11.58 |
| | $\ell_1$-sparse | $99.32_{\pm 0.04}(0.65)$ | $31.71_{\pm 0.52}(7.02)$ | $63.33_{\pm 0.32}(1.44)$ | $46.49_{\pm 0.82}(15.26)$ | 6.09 | 14.47 |
| | SalUn | $99.18_{\pm 0.13}(0.79)$ | $38.01_{\pm 2.43}(0.72)$ | $54.78_{\pm 0.52}(7.11)$ | $69.24_{\pm 1.92}(7.49)$ | 4.03 | 5.10 |
| | NoT | $99.89_{\pm 0.02}(0.08)$ | $20.29_{\pm 1.93}(18.44)$ | $61.82_{\pm 0.29}(0.07)$ | $43.55_{\pm 1.36}(18.20)$ | 9.20 | 8.68 |
| | CoUn | $99.91_{\pm 0.03}(0.06)$ | $36.81_{\pm 1.08}(1.92)$ | $56.49_{\pm 0.55}(5.40)$ | $53.92_{\pm 0.42}(7.83)$ | 3.80 | 19.29 |

**Evaluation Metrics.** Following [8, 9, 10], we evaluate the unlearning effectiveness and efficiency of `CoUn` using the following empirical metrics: ❶ *Retain Accuracy (RA)*: Accuracy of the unlearned model $\theta_u$ on retain data $\mathcal{D}_r$. ❷ *Unlearn Accuracy (UA)*: Measured as $1-FA$, where Forget Accuracy (**FA**) is the accuracy of $\theta_u$ on forget data $\mathcal{D}_u$. ❸ *Test Accuracy (TA)*: Generalization performance of $\theta_u$ on test data. ❹ *Membership Inference Attack (MIA)*: The efficacy of unlearning, evaluated using a confidence-based MIA predictor [9, 11, 10, 13] applied to $\theta_u$ on $\mathcal{D}_u$. The MIA success rate reflects how effectively forget data is excluded from training. ❺ *Computation Cost*: Efficiency measured by the number of floating-point operations (FLOPs) required to generate $\theta_u$. The model's **utility** is assessed using RA and TA, while its **forget quality** is evaluated using both UA and MIA metrics. A higher value in any individual metric (e.g., RA, UA, TA, or MIA) does not necessarily indicate better performance. An effective unlearning method minimizes the performance gap with the gold-standard Retrain model. Therefore, the **performance** (i.e., **unlearning effectiveness**) of an MU method is measured by the *average gap*:

$$Avg.\ Gap = \frac{1}{4}\big(|RA - RA^*| + |UA - UA^*| + |TA - TA^*| + |MIA - MIA^*|\big), \quad (7)$$

where $^*$ denotes metrics for the Retrain model.

**Implementation Details.** *For training the Original and Retrain models*: we follow prior work [8, 9, 10] by using an initial learning rate of 0.1, which is reduced by a factor of 10 at 50% and 75% of the total 182 training epochs. The batch size is set to 256. *For unlearning*: all MU methods are applied to $\theta_o$ for 50 epochs, using a cosine learning rate scheduler with a minimum learning rate of $10^{-4}$. All reported results are averaged over 10 trials. Additional details can be found in Appendix E.2.

Table 3: **Performance comparison of** `CoUn` **to the baseline methods** with **50% random data removal**. The gap ($\Delta$) and the (best) average gap between each method and the Retrain model are reported.

| Dataset & Model | Method | Accuracy (%) | | | Efficacy (%) | Avg. Gap ↓ | Comp. Cost (PFLOPs) ↓ |
|---|---|---|---|---|---|---|---|
| | | Retain ($\Delta$ ↓) | Unlearn ($\Delta$ ↓) | Test ($\Delta$ ↓) | MIA ($\Delta$ ↓) | | |
| CIFAR-10 ResNet-18 | Retrain | $100.00_{\pm 0.00}(0.00)$ | $7.29_{\pm 0.36}(0.00)$ | $92.28_{\pm 0.23}(0.00)$ | $17.33_{\pm 0.65}(0.00)$ | 0.00 | 15.24 |
| | FT | $99.38_{\pm 0.24}(0.62)$ | $6.32_{\pm 0.41}(0.97)$ | $91.91_{\pm 0.41}(0.37)$ | $12.64_{\pm 0.51}(4.69)$ | 1.66 | 2.51 |
| | NegGrad+ | $100.00_{\pm 0.00}(0.00)$ | $4.06_{\pm 0.20}(0.75)$ | $93.81_{\pm 0.23}(0.86)$ | $9.05_{\pm 0.22}(1.97)$ | 0.90 | 5.58 |
| | $\ell_1$-sparse | $99.77_{\pm 0.03}(0.23)$ | $9.02_{\pm 0.16}(1.73)$ | $90.66_{\pm 0.24}(1.62)$ | $16.05_{\pm 0.29}(1.28)$ | 1.22 | 4.19 |
| | SalUn | $98.70_{\pm 0.29}(1.30)$ | $3.74_{\pm 0.32}(3.55)$ | $92.37_{\pm 0.36}(0.09)$ | $16.40_{\pm 1.14}(0.93)$ | 1.47 | 7.82 |
| | NoT | $99.98_{\pm 0.01}(0.02)$ | $5.95_{\pm 0.18}(1.34)$ | $92.84_{\pm 0.18}(0.56)$ | $13.90_{\pm 0.37}(3.43)$ | 1.34 | 3.35 |
| | CoUn | $99.97_{\pm 0.03}(0.03)$ | $6.19_{\pm 0.30}(1.10)$ | $92.36_{\pm 0.26}(0.08)$ | $16.94_{\pm 0.48}(0.39)$ | 0.40 | 3.35 |
| CIFAR-100 ResNet-18 | Retrain | $99.98_{\pm 0.01}(0.00)$ | $31.41_{\pm 0.40}(0.00)$ | $68.41_{\pm 0.34}(0.00)$ | $58.35_{\pm 0.53}(0.00)$ | 0.00 | 15.24 |
| | FT | $99.98_{\pm 0.01}(0.00)$ | $17.36_{\pm 0.19}(14.05)$ | $74.16_{\pm 0.39}(5.75)$ | $50.60_{\pm 0.42}(7.75)$ | 6.89 | 4.19 |
| | NegGrad+ | $99.98_{\pm 0.01}(0.00)$ | $26.32_{\pm 0.21}(5.09)$ | $71.98_{\pm 0.30}(3.57)$ | $52.32_{\pm 0.36}(6.03)$ | 3.67 | 5.36 |
| | $\ell_1$-sparse | $99.94_{\pm 0.02}(0.04)$ | $32.26_{\pm 0.23}(0.85)$ | $67.66_{\pm 0.35}(0.75)$ | $51.54_{\pm 0.29}(6.81)$ | 2.11 | 4.19 |
| | SalUn | $95.61_{\pm 0.61}(4.37)$ | $25.43_{\pm 1.32}(5.98)$ | $60.35_{\pm 0.82}(8.06)$ | $57.14_{\pm 1.06}(1.21)$ | 4.91 | 1.95 |
| | NoT | $98.64_{\pm 0.43}(1.34)$ | $26.43_{\pm 0.75}(4.98)$ | $67.97_{\pm 0.83}(0.44)$ | $43.82_{\pm 0.60}(14.53)$ | 5.32 | 2.01 |
| | CoUn | $99.98_{\pm 0.01}(0.00)$ | $31.43_{\pm 1.75}(0.02)$ | $65.60_{\pm 0.71}(2.81)$ | $55.99_{\pm 1.18}(2.36)$ | 1.30 | 5.58 |
| TinyImageNet ResNet-18 | Retrain | $99.99_{\pm 0.00}(0.00)$ | $43.01_{\pm 0.20}(0.00)$ | $57.28_{\pm 0.43}(0.00)$ | $71.22_{\pm 0.17}(0.00)$ | 0.00 | 121.93 |
| | FT | $99.99_{\pm 0.00}(0.00)$ | $36.78_{\pm 0.18}(6.23)$ | $60.59_{\pm 0.38}(3.31)$ | $66.28_{\pm 0.20}(4.94)$ | 3.62 | 33.50 |
| | NegGrad+ | $99.99_{\pm 0.00}(0.00)$ | $47.62_{\pm 0.25}(4.61)$ | $58.85_{\pm 0.32}(1.57)$ | $66.43_{\pm 0.33}(4.79)$ | 2.74 | 33.50 |
| | $\ell_1$-sparse | $99.99_{\pm 0.00}(0.00)$ | $38.83_{\pm 0.21}(4.18)$ | $60.25_{\pm 0.30}(2.97)$ | $65.82_{\pm 0.21}(5.40)$ | 3.14 | 33.50 |
| | SalUn | $93.59_{\pm 0.55}(6.40)$ | $44.74_{\pm 1.11}(1.73)$ | $45.53_{\pm 0.91}(11.75)$ | $70.41_{\pm 1.05}(0.81)$ | 5.17 | 25.01 |
| | NoT | $99.99_{\pm 0.00}(0.00)$ | $40.94_{\pm 0.43}(2.07)$ | $58.27_{\pm 0.39}(0.99)$ | $66.23_{\pm 0.36}(4.99)$ | 2.01 | 33.50 |
| | CoUn | $99.98_{\pm 0.01}(0.01)$ | $43.21_{\pm 1.57}(0.20)$ | $55.75_{\pm 1.34}(1.53)$ | $66.59_{\pm 0.41}(4.63)$ | 1.59 | 44.66 |
| CIFAR-100 VGG-16 | Retrain | $99.65_{\pm 0.18}(0.00)$ | $42.85_{\pm 0.54}(0.00)$ | $57.70_{\pm 0.47}(0.00)$ | $50.19_{\pm 0.93}(0.00)$ | 0.00 | 8.67 |
| | FT | $97.71_{\pm 0.25}(1.94)$ | $29.82_{\pm 0.58}(13.03)$ | $63.72_{\pm 0.43}(6.02)$ | $39.98_{\pm 0.62}(10.21)$ | 7.80 | 1.43 |
| | NegGrad+ | $95.54_{\pm 0.56}(4.11)$ | $43.42_{\pm 0.38}(0.57)$ | $58.52_{\pm 0.44}(0.82)$ | $43.51_{\pm 0.36}(6.68)$ | 3.04 | 3.18 |
| | $\ell_1$-sparse | $98.25_{\pm 1.53}(1.40)$ | $34.24_{\pm 1.87}(8.61)$ | $62.76_{\pm 1.65}(5.06)$ | $42.12_{\pm 0.55}(8.07)$ | 5.79 | 1.91 |
| | SalUn | $91.98_{\pm 0.75}(7.67)$ | $37.60_{\pm 3.52}(5.25)$ | $57.30_{\pm 0.83}(0.40)$ | $53.84_{\pm 5.95}(3.65)$ | 4.24 | 2.45 |
| | NoT | $94.23_{\pm 7.94}(5.42)$ | $34.64_{\pm 7.05}(8.21)$ | $61.58_{\pm 4.67}(3.88)$ | $39.84_{\pm 1.62}(10.35)$ | 6.96 | 2.38 |
| | CoUn | $99.88_{\pm 0.02}(0.23)$ | $42.37_{\pm 0.80}(0.48)$ | $55.19_{\pm 0.68}(2.51)$ | $50.00_{\pm 0.68}(0.19)$ | 0.85 | 3.18 |
| CIFAR-100 ViT | Retrain | $99.98_{\pm 0.01}(0.00)$ | $48.07_{\pm 0.33}(0.00)$ | $52.40_{\pm 0.58}(0.00)$ | $69.54_{\pm 0.29}(0.00)$ | 0.00 | 48.35 |
| | FT | $98.71_{\pm 0.30}(1.27)$ | $10.91_{\pm 0.96}(37.16)$ | $56.79_{\pm 0.73}(4.39)$ | $28.18_{\pm 0.93}(41.36)$ | 21.05 | 1.61 |
| | NegGrad+ | $99.30_{\pm 0.13}(0.68)$ | $45.35_{\pm 0.48}(2.72)$ | $50.82_{\pm 0.47}(1.58)$ | $55.07_{\pm 0.33}(14.47)$ | 4.86 | 6.45 |
| | $\ell_1$-sparse | $71.18_{\pm 0.57}(28.80)$ | $47.30_{\pm 0.26}(0.77)$ | $53.32_{\pm 0.52}(0.92)$ | $44.22_{\pm 2.98}(25.32)$ | 13.95 | 8.06 |
| | SalUn | $98.93_{\pm 0.29}(1.05)$ | $45.64_{\pm 2.99}(2.43)$ | $39.46_{\pm 0.82}(12.94)$ | $76.49_{\pm 1.92}(6.95)$ | 5.84 | 12.03 |
| | NoT | $97.86_{\pm 1.71}(2.12)$ | $31.81_{\pm 2.05}(16.26)$ | $55.51_{\pm 1.15}(3.11)$ | $48.85_{\pm 2.48}(20.69)$ | 10.55 | 3.22 |
| | CoUn | $99.71_{\pm 0.62}(0.27)$ | $45.95_{\pm 3.70}(2.12)$ | $49.45_{\pm 2.15}(2.95)$ | $59.24_{\pm 1.48}(10.30)$ | 3.91 | 10.74 |

## 4.2 Results

**Comparison with Baselines.** We begin by presenting results for the *10% random forget data ratio*. As shown in Table 2, `CoUn` consistently outperforms state-of-the-art baselines, with computational costs that are either comparable or slightly higher. Key observations from Table 2 include the following: FT exhibits the highest average gap, indicating weakest performance. SalUn shows limited performance compared to other baselines, as the random labeling of forget data reduces the model's utility. While NegGrad+, $\ell_1$-sparse and NoT achieve the most competitive results, `CoUn` demonstrates superior performance across different datasets and model architectures. For example, using ResNet-18, `CoUn` outperforms the best baseline by 16.7% on CIFAR-10, 15.8% on CIFAR-100, and 6.3% on TinyImageNet. Similarly, with VGG-16 and ViT on CIFAR-100, `CoUn` achieves performance improvements of 53.4% and 4.5%, respectively. Even under a *50% random forget data ratio*, `CoUn` still outperforms the baselines (Table 3). Comparisons with *additional baselines* and *class-wise forgetting* are provided in Appendix F.1 and Appendix F.2, respectively.

**Integration of `CoUn`'s CL Module into Baselines.** To evaluate whether our proposed CL module empowers baseline methods, we conduct experiments integrating `CoUn`'s CL module with competitive baselines. Figure 4 presents the average gap comparisons, with percentage improvements, across various datasets and model architectures for 10% and 50% forget data ratios. Detailed results can be found in Appendix F.3. Our results demonstrate a substantial performance boost achieved by incorporating our CL module into the baselines. For example, using CIFAR-10 with ResNet-18, integrating our CL module with NegGrad+, $\ell_1$-sparse and NoT results in percentage improvements of 44.8%, 42.1%, and 43.3%, respectively, under a 10% forget data ratio; while for a 50% forget ratio, improvements of 50.0%, 87.7%, and 73.1% are achieved. Although the CL module introduces a slight increase in computational cost due to the additional model inference required for obtaining representations of the second sample view, Figure 5 shows that the performance enhancements remain significant even when the computation budgets are matched.

**Sequential Unlearning.** Figure 6 presents results for scenarios where 10% of random data is sequentially removed every 10 epochs, up to 50 epochs. CoUn consistently outperforms baseline methods across all five stages, with varying forget ratios. Additionally, integrating CoUn's CL module into baseline methods further empowers their unlearning effectiveness.

## 4.3 Ablation Study

All ablation experiments are conducted using CIFAR-100, ResNet-18, and a forget ratio of 50%.

**Effect of Scaling Factor.** The scaling factor $\lambda$, defined in Equation (4), controls the relative contribution of the CE and CL losses. Figure 7 demonstrates the substantial impact of $\lambda$ on CoUn's performance. Improper tuning can lead to suboptimal results, emphasizing the importance of careful hyperparameter selection. For example, a high $\lambda$ reduces the influence of supervised learning in the objective, causing retain representations to be less constrained within their respective clusters and more susceptible to cluster collisions. This, in turn, degrades model utility and compromises unlearning effectiveness.

**Effect of CL Temperature.** Similar to SimCLR [20], we investigate the influence of the CL temperature $\tau$, defined in Equation (1), on unlearning effectiveness. Figure 8 demonstrates that decreasing $\tau$ generally improves the effectiveness of CL, and reduces the average gap with the Retrain model. Nevertheless, excessively low $\tau$ values can negatively impact performance. As shown in Figure 8, optimal performance is observed at $\tau = 0.1$, with performance deteriorating as $\tau$ deviates from this value. In line with previously reported findings from SimCLR, the results confirm that the best performance emerges at low $\tau$, whereas performance gradually declines as $\tau$ increases beyond the optimal value.

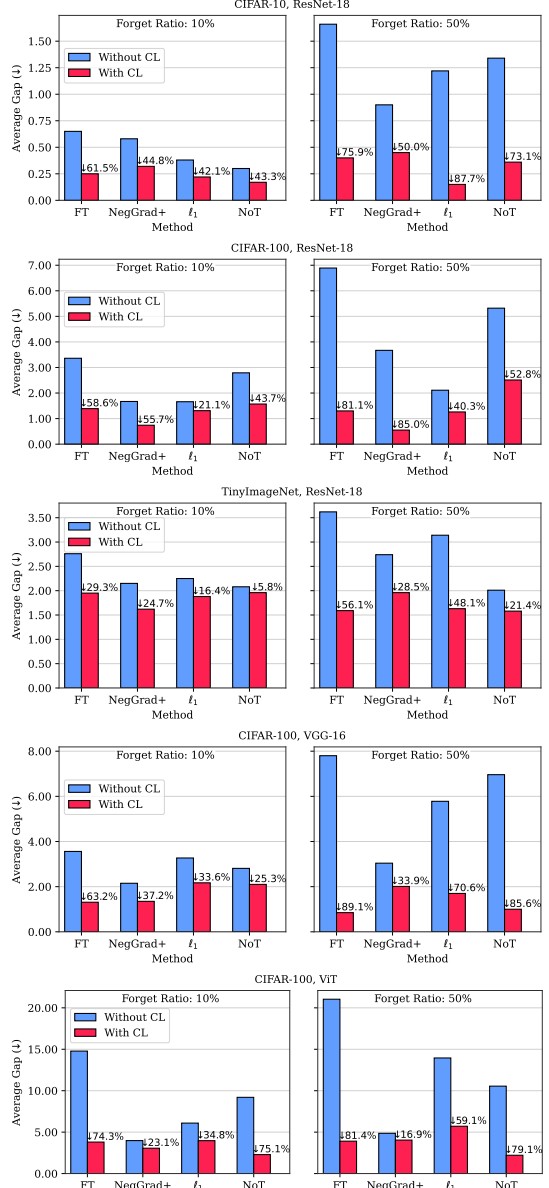

Figure 4: **Percentage improvement from integrating CoUn's CL module into baseline methods**. Incorporating our CL module consistently improves baseline unlearning performance compared to the original MU methods (without CL). The performance improvements further increase with a 50% forget ratio.

**Effect of CL Transformation Distribution.** To isolate the impact of CL transformation distributions $\mathcal{T}_{\mathrm{CL}}$, we fix the supervised learning transformation $\mathcal{T}_{\mathrm{CE}}$ to CHN, consistent with the transformation used in the Retrain model. The details of the augmentation operations C, H, J, G, and N are provided in Appendix E.3. Figure 9 demonstrates that employing stronger transformations for CL (e.g., CHJGN), compared to those used for supervised learning, can degrade performance. This degradation arises primarily from the formation of tighter clustering of forget representations, along with slower convergence caused by the additional complexity introduced by stronger transformations. On the other hand, overly simple transformations (e.g., CN) may not sufficiently adjust the forget representations, which in turn result in suboptimal performance. Our experiments show that the best performance is achieved when $\mathcal{T} = \mathcal{T}_{\mathrm{CE}} = \mathcal{T}_{\mathrm{CL}}$, i.e., when $\mathcal{T} = \mathrm{CHN}$. This configuration also enables shared use of representations from a single augmented image between CL and supervised learning, thereby

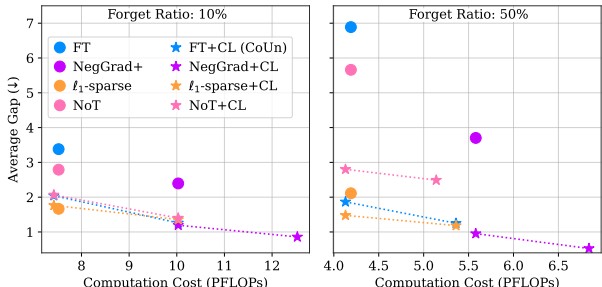
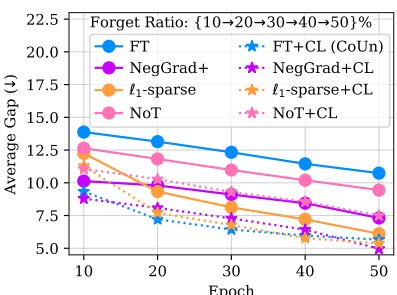

Figure 5: **Performance comparison of MU methods** on CIFAR-100 with ResNet-18, where 10% (*left*) and 50% (*right*) of training data are randomly selected as forget data. The best performance of each method is reported. `CoUn` outperforms all baselines, and integrating its CL module empowers baseline performance. Although CL increases computational cost, the performance improvement persists even with the same computational budget.

Figure 6: **Sequential data removal.** Experiments with CIFAR-100, ResNet-18, and up to 50% random forget data (10% of data is removed every 10 epochs). `CoUn` consistently outperforms baselines, and can further empower baselines' performance when `CoUn`'s CL module is integrated into them.

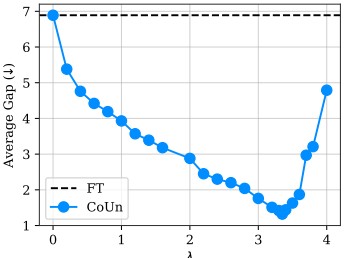
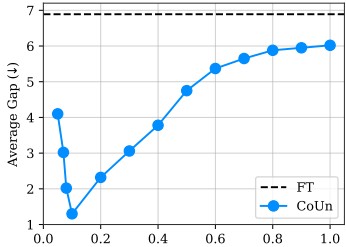
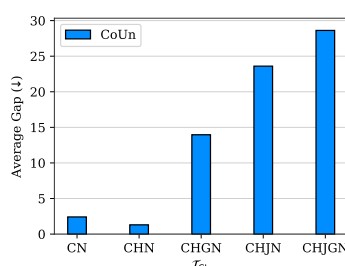

Figure 7: **Effect of scaling constant $\lambda$.** Properly tuning $\lambda$ in Equation (4) is essential for optimizing `CoUn`'s performance.

Figure 8: **Effect of CL temperature $\tau$.** Properly tuning $\tau$ in Equation (1) is essential for optimizing `CoUn`'s performance.

Figure 9: **Effect of CL transformation $\mathcal{T}_{CL}$.** Strong $\mathcal{T}_{CL}$ degrades performance, while simple $\mathcal{T}_{CL}$ fails to sufficiently push representations.

reducing computational cost. The impact of strong versus simple CL transformations on forget representations is further illustrated in Appendix F.4.

**Effect of Batch Size**   Batch size impacts the performance of `CoUn`. Figure 10 presents results for varying batch sizes, with the batch size for the Retrain model fixed at 256. Our findings indicate that the best performance for `CoUn` is achieved with a batch size of 256, which matches the batch size used for the Retrain model.

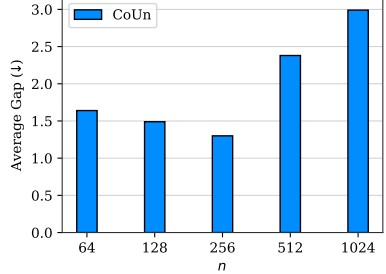

Figure 10: **Effect of batch size $n$.** Different $n$ for `CoUn`, results in varying performance. Retrain batch size is set to 256.

# 5   Conclusion

We presented `CoUn`, a novel CL-based MU framework that enables effective unlearning by adjusting learned data representations based on semantic similarity. `CoUn` applies a CL module on retain data to adjust their representations and leverages the cluster collision issue to promote cluster overlap. Due to semantic similarity between retain and forget samples, forget representations are indirectly influenced in the same manner; thereby enhancing forget quality. To preserve utility, `CoUn` applies supervised learning to retain data to mitigate cluster collision for retain representations. Our results showed that `CoUn` consistently outperforms state-of-the-art MU baselines, and that integrating its CL module into existing baselines empowers their unlearning effectiveness.

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

We provide more details and results about our work in the appendices. Here are the contents:

# A   Related Work: Further Details

**Contrastive Learning.**   Zhang et al. [33] applies *supervised CL* [41] to push forget samples away from retain samples of the same cluster and pull them closer to retain samples of different clusters. Essentially, [33] pushes representations away from positive samples and toward negative ones. This approach requires that forget and retain samples from the same cluster to be included in each batch. Moreover, under this definition there are no positive samples in class-wise unlearning, thus [33] modifies the objective to only pull forget samples toward retain samples from different clusters. However, this approach aims to push forget samples outside their clusters, potentially harming model utility. As shown in Figure 1, the goal of unlearning is not to misclassify forget samples as some can be correctly classified to maintain model performance. Additionally, [33] requires access to forget data.

In contrast, CoUn follows the way how Retrain model classifies forget data, which is based on semantic similarity. CoUn utilizes *self-supervised CL* [18, 19, 20, 29, 30, 31, 32] to achieve the same goal. Self-supervised CL uses augmentations to generate positive views, instead of using samples from different clusters as the positive views. The use of augmented samples as positives and the remaining as negatives allows three advantages: ❶ access to class labels is not required during CL, ❷ we do not need to guarantee that samples from different clusters need to exist in the batch, and ❸ it does not force samples out of their original clusters. Lastly, CoUn does not require access to forget data.

# B   CoUn Algorithm

## B.1   Pseudo-Code

Algorithm 1 details our proposed unlearning method, which leverages CL and supervised learning.

---
**Algorithm 1** CoUn Algorithm
---
**Input**: Original model $\boldsymbol{\theta}_o$, transformation distribution $\mathcal{T}$, and retain data $\mathcal{D}_r$
**Hyper-parameter**: Learning rate $\eta$, temperature $\tau$, and scaling factor $\lambda$
**Output**: Unlearned model $\boldsymbol{\theta}_u$

1:  $\boldsymbol{\theta}_u \leftarrow \boldsymbol{\theta}_o$
2:  **for** epoch $e = 1, 2, \ldots, E$ **do**
3:      **for** each batch $(\boldsymbol{I}, \boldsymbol{Y}) \in \mathcal{D}_r$ **do**
4:          Sample transformations $t, t' \sim \mathcal{T}$
5:          $\boldsymbol{X}, \boldsymbol{X}' = t(\boldsymbol{I}), t'(\boldsymbol{I})$
6:          $\boldsymbol{Z}, \boldsymbol{Z}' = f_{\boldsymbol{\theta}_u}(\boldsymbol{X}), f_{\boldsymbol{\theta}_u}(\boldsymbol{X}')$
7:          $\hat{\boldsymbol{Y}} = h_{\boldsymbol{\theta}_u}(\boldsymbol{Z})$
8:          $\mathcal{L}_{\mathrm{CL}}$ is obtained from Equation (2) using $\boldsymbol{Z}, \boldsymbol{Z}'$
9:          $\mathcal{L}_{\mathrm{CE}}$ is obtained from Equation (3) using $\boldsymbol{Y}, \hat{\boldsymbol{Y}}$
10:          $\boldsymbol{\theta}_u \leftarrow \boldsymbol{\theta}_u - \eta \nabla_{\boldsymbol{\theta}_u} (\mathcal{L}_{\mathrm{CE}} + \lambda \mathcal{L}_{\mathrm{CL}})$
11:      **end for**
12:  **end for**
13:  **return** $\boldsymbol{\theta}_u$
---

## B.2 PyTorch Code

This section provides the PyTorch implementation of `CoUn`.

```python
import torch
from torch import nn
from torch.nn import functional as F

features = None
hook_fn = lambda module, _, output: globals().__setitem__('features',
    output)

def coun(model, layer, optimizer, retain_loader, transform,
    lambda_scale, temp):
    '''Apply contrastive learning for unlearning using only retain
    data
    Args:
        model: The original model that needs to be unlearned
        layer: The penultimate layer for extracting embeddings
        optimizer: The optimizer to train the model
        retain_loader: The dataloader containing retain data
        transform: The transformation to be applied to input images
        lambda_scale: A scaling constant for the CL loss
        temp: The temperature to be applied in the CL loss
    returns 'unlearned model' '''

    _ = layer.register_forward_hook(hook_fn) # Attach hook at
    penultimate layer
    for images, targets in retain_loader: # Load retain data
        batch_size = int(images.shape[0])

        # Create two views of images
        images1, images2 = transform(images), transform(images)

        # Get model outputs and extract embeddings for images1
        outputs = model(images1)
        features1 = features.view(batch_size, -1)

        # Extract embeddings for images2
        _ = model(images2)
        features2 = features.view(batch_size, -1)

        # Compute supervised learning loss for images1
        supervised_loss = nn.CrossEntropyLoss()(outputs, targets)

        # Compute contrastive learning using the two views of
    embeddings
        target = torch.arange(batch_size).unsqueeze(0)
        intra_mask = (torch.eq(target, target.T).float())

        cos_sim_ij = F.cosine_similarity(features1[:,None,:],
    features2[None,:,:], dim=-1)
        cos_sim_ij = torch.div(cos_sim_ij, temp)
        log_prob_ij = cos_sim_ij - torch.log((torch.exp(cos_sim_ij)).
    sum(1, keepdim=True))
        mean_log_prob_pos_ij = (intra_mask * log_prob_ij).sum(1) /
    intra_mask.sum(1)

        cos_sim_ji = F.cosine_similarity(features2[:,None,:],
    features1[None,:,:], dim=-1)
        cos_sim_ji = torch.div(cos_sim_ji, temp)
        log_prob_ji = cos_sim_ji - torch.log((torch.exp(cos_sim_ji)).
    sum(1, keepdim=True))
        mean_log_prob_pos_ji = (intra_mask * log_prob_ji).sum(1) /
    intra_mask.sum(1)
```

```
51
52        constrastive_loss = - (mean_log_prob_pos_ij.mean() +
      mean_log_prob_pos_ji.mean())
53
54        loss = supervised_loss + lambda_scale*constrastive_loss
55
56        optimizer.zero_grad()
57        loss.backward()
58        optimizer.step()
59
60    return model
```

## C  Further Details on Empirical Analysis

**Representation Space of Original Model.**  Figure 11 illustrates the representation space of the Original model trained on CIFAR-10 using ResNet-18. Since it is trained on both retain and forget data, the model achieves perfect accuracy on both.

**Statistical Comparisons.**  We provide a statistical comparison between forget representations and retain clusters, we grouped the forget samples by their true class labels and, for each group, computed the average Euclidean distance (L2) from its samples to all retain class centroids. This yielded a per-class distance profile showing how far forget representations lie from each retain cluster. To enable comparison across different models, we then applied normalization

Table 4: **L2 distances of forget 'truck' to retain centroids.** The most semantically similar clusters to 'truck' samples are presented. Experiments conducted using CIFAR-10 and ResNet-18. The difference ($\Delta$) and the (best) average difference between each method and Retrain are reported.

| Forgetting Scenario | Method | L2 - ($\Delta \downarrow$) | | | Avg. Diff. $\downarrow$ |
|---|---|---|---|---|---|
| | | **Automobile** | **Airplane** | **Ship** | |
| Class ('truck') | Original | 0.93 | 0.97 | 0.96 | - |
| | Retrain | 0.90 (0.00) | 0.96 (0.00) | 0.95 (0.00) | 0.00 |
| | FT | 0.86 (0.04) | 0.94 (0.02) | 0.91 (0.04) | 0.033 |
| | CoUn | 0.87 (0.03) | 0.96 (0.00) | 0.93 (0.02) | 0.017 |

on each group's averaged distances. Table 4 summarizes the results for CIFAR-10, ResNet-18, and the statistics for 'truck' forget samples in a 10% random forgetting (same setup as Table 1). The findings show that forget representations in `CoUn` are consistently closer to semantically similar retain clusters, and more importantly, `CoUn` achieves distance statistics that are closer to those of the Retrain model compared to other baselines. The smaller the distance means higher semantic similarity. From Table 4, we can see that 'truck' samples have the highest semantic similarity with 'automobile'.

**Prediction-Level Results.**  Tables 5 and 6 present additional prediction-level results for a single forget class across different baselines and datasets. The Original model has 100% accuracy on the forget 'truck' samples since these samples are part of its training data. Furthermore, baselines based on label manipulation or weight perturbation produce predictions somewhat similar to Retrain, but their misclassifications are less concentrated on semantically related classes. By comparison, `CoUn` more effectively redirects forget samples toward semantically similar clusters, thereby yielding prediction distributions that are closer to those of the Retrain model.

## D  Proof of Lemma 1

We have

$$\|f_{\boldsymbol{\theta}_u}(\boldsymbol{x}) - f_{\boldsymbol{\theta}_u}(\boldsymbol{x}')\| \leq L \|\boldsymbol{x} - \boldsymbol{x}'\|, \tag{8}$$

$$\sup_{\substack{\boldsymbol{x}=t(\boldsymbol{i}),\, \boldsymbol{x}'\in t'(\boldsymbol{i}) \\ t,\, t' \sim \mathcal{T}}} \|f_{\boldsymbol{\theta}_u}(\boldsymbol{x}) - f_{\boldsymbol{\theta}_u}(\boldsymbol{x}')\| \leq L \sup_{\substack{\boldsymbol{x}=t(\boldsymbol{i}),\, \boldsymbol{x}'\in t'(\boldsymbol{i}) \\ t,\, t' \sim \mathcal{T}}} \|\boldsymbol{x} - \boldsymbol{x}'\|, \tag{9}$$

where inequality (8) is obtained due to the L-Lipschitz assumption for feature extractor $f_{\boldsymbol{\theta}_u}$ and inequality (9) is obtained by taking supremum from both sides of inequality (8).

Training $f_{\boldsymbol{\theta}_u}$ on the retain data tends to converge to flat minima, i.e., regions in parameter space where the loss landscape is broad and has low curvature [42]. Consequently, flat minima correspond

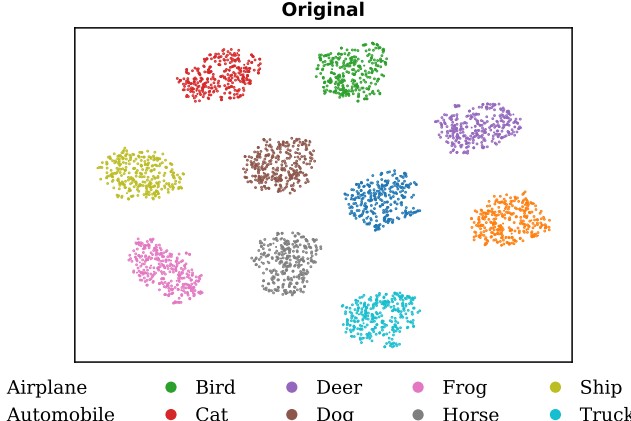

**Original**

| | | | | |
|---|---|---|---|---|
| ● Airplane | ● Bird | ● Deer | ● Frog | ● Ship |
| ● Automobile | ● Cat | ● Dog | ● Horse | ● Truck |

Figure 11: **Representation space of the Original model**. The Original model is trained on the entire CIFAR-10 training data (i.e., union of retain and forget data) using ResNet-18. There are no misclassifications for either retain or forget samples since the model was trained on them. A single visualization of the Original's model representation space is shown for both class-wise and random scenarios, as this model serves as the checkpoint for both scenarios. Dots denote training samples, where each being correctly clustered into the corresponding class.

to functions whose outputs vary only gently under small perturbations around the training points (i.e., they have a low local Lipschitz constant at those points). However, there is no guarantee that, at these flat minima, the loss landscape remains flat with respect to the forget data. Therefore, in general, after convergence we have

$$L_r \leq L_u \leq L, \tag{10}$$

where $L_r$ and $L_u$ denote the Lipschitz constants evaluated on the retain and forget data, respectively.

Based on Equation (5) and inequality (9), we have

$$R_r[\epsilon] = P\left[ \boldsymbol{i} \in \left( \bigcup_{k=1}^{K} C_k \right) \cap \mathcal{D}_r \,\middle|\, \sup_{\substack{\boldsymbol{x}=t(\boldsymbol{i}),\, \boldsymbol{x}'\in t'(\boldsymbol{i}) \\ t,\, t'\sim\mathcal{T}}} \|\boldsymbol{x} - \boldsymbol{x}'\| > \epsilon/L_r \right], \tag{11}$$

and

$$R_u[\epsilon] = P\left[ \boldsymbol{i} \in \left( \bigcup_{k=1}^{K} C_k \right) \cap \mathcal{D}_u \,\middle|\, \sup_{\substack{\boldsymbol{x}=t(\boldsymbol{i}),\, \boldsymbol{x}'\in t'(\boldsymbol{i}) \\ t,\, t'\sim\mathcal{T}}} \|\boldsymbol{x} - \boldsymbol{x}'\| > \epsilon/L_u \right]. \tag{12}$$

In particular, Equation (11) computes the probability that, for images in the retain data, $\sup_{\substack{\boldsymbol{x}=t(\boldsymbol{i}),\, \boldsymbol{x}'\in t'(\boldsymbol{i}) \\ t,\, t'\sim\mathcal{T}}} \|\boldsymbol{x} - \boldsymbol{x}'\|$ exceeds $\epsilon/L_r$, while Equation (12) computes the probability that, for images in the forget data, $\sup_{\substack{\boldsymbol{x}=t(\boldsymbol{i}),\, \boldsymbol{x}'\in t'(\boldsymbol{i}) \\ t,\, t'\sim\mathcal{T}}} \|\boldsymbol{x} - \boldsymbol{x}'\|$ exceeds $\epsilon/L_u$. Considering inequality (10), we have $\epsilon/L_u \leq \epsilon/L_r$. Since in random forgetting the samples are drawn I.I.D., we have $R_r[\epsilon] \leq R_u[\epsilon]$. This follows because the threshold $\epsilon/L_u$ used to compute $R_u[\epsilon]$ is lower than the threshold $\epsilon/L_r$ used to compute $R_r[\epsilon]$.

# E    Further Implementation Details

## E.1    Model Architectures

For VGG-16 [39], we use a 1024-dimensional encoder head. For the ViT [40] model, we adopt a patch size of $4 \times 4$, an embedding dimension of 512, an MLP hidden dimension of 1024, 12 attention heads of size 64, and a depth of 6 transformer layers. Both dropout and embedding dropout are set to 0.1.

Table 5: **Predictions of forget 'truck' samples based on most semantically similar classes**. The experiments are conducted using CIFAR-10 and ResNet-18. The difference ($\Delta$) and the (best) average difference between each method and Retrain are reported.

| Forgetting Scenario | Method | Predictions (%) - ($\Delta \downarrow$) | | | | Avg. Diff. $\downarrow$ |
|---|---|---|---|---|---|---|
| | | **Truck** | **Automobile** | **Airplane** | **Ship** | |
| Class ('truck') | Original | 100.00 (0.00) | 0.00 (0.00) | 0.00 (0.00) | 0.00 (0.00) | 0.00 |
| | Retrain | 0.00 (0.00) | 69.32 (0.00) | 13.47 (0.00) | 12.60 (0.00) | 0.00 |
| | FT | 0.00 (0.00) | 70.29 (0.97) | 12.38 (1.09) | 13.12 (0.52) | 0.65 |
| | NegGrad+ | 0.00 (0.00) | 43.56 (25.76) | 16.03 (2.56) | 17.58 (4.98) | 8.32 |
| | $\ell_1$-sparse | 0.00 (0.00) | 65.70 (3.62) | 9.84 (3.63) | 18.25 (5.65) | 3.23 |
| | SalUn | 0.00 (0.00) | 66.05 (3.27) | 16.67 (3.20) | 16.71 (4.11) | 2.65 |
| | NoT | 0.00 (0.00) | 68.12 (1.20) | 12.43 (1.04) | 15.03 (2.43) | 1.17 |
| | CoUn | 0.00 (0.00) | 69.60 (0.28) | 13.96 (0.49) | 13.13 (0.53) | 0.33 |
| Random (10%) | Original | 100.00 (0.00) | 0.00 (0.00) | 0.00 (0.00) | 0.00 (0.00) | 0.00 |
| | Retrain | 97.42 (0.00) | 1.23 (0.00) | 0.38 (0.00) | 0.40 (0.00) | 0.00 |
| | FT | 98.12 (0.70) | 0.75 (0.48) | 0.36 (0.02) | 0.32 (0.08) | 0.32 |
| | NegGrad+ | 97.86 (0.44) | 0.97 (0.26) | 0.42 (0.04) | 0.30 (0.10) | 0.21 |
| | $\ell_1$-sparse | 98.06 (0.64) | 0.85 (0.38) | 0.30 (0.08) | 0.38 (0.02) | 0.28 |
| | SalUn | 96.76 (0.66) | 0.88 (0.35) | 0.34 (0.04) | 0.34 (0.06) | 0.28 |
| | NoT | 97.98 (0.56) | 0.89 (0.34) | 0.40 (0.02) | 0.32 (0.08) | 0.25 |
| | CoUn | 97.84 (0.42) | 0.99 (0.24) | 0.32 (0.06) | 0.42 (0.02) | 0.19 |

Table 6: **Predictions of forget 'man' samples based on most semantically similar classes**. The experiment is conducted using CIFAR-100 and VGG-16. The difference ($\Delta$) and the (best) average difference between each method and Retrain are reported.

| Forgetting Scenario | Method | Predictions (%) - ($\Delta \downarrow$) | | | | Avg. Diff. $\downarrow$ |
|---|---|---|---|---|---|---|
| | | **Man** | **Woman** | **Boy** | **Baby** | |
| Random (10%) | Original | 100.00 (0.00) | 0.00 (0.00) | 0.00 (0.00) | 0.00 (0.00) | 0.00 |
| | Retrain | 49.53 (0.00) | 15.18 (0.00) | 10.06 (0.00) | 3.42 (0.00) | 0.00 |
| | FT | 60.53 (11.00) | 9.11 (6.07) | 9.68 (0.38) | 3.61 (0.19) | 4.41 |
| | NegGrad+ | 41.94 (7.59) | 13.47 (1.71) | 12.71 (2.65) | 4.74 (1.32) | 3.32 |
| | $\ell_1$-sparse | 57.69 (8.16) | 13.47 (1.71) | 10.06 (0.00) | 2.85 (0.57) | 2.61 |
| | SalUn | 55.16 (5.63) | 13.52 (1.66) | 9.68 (0.38) | 2.63 (0.79) | 2.12 |
| | NoT | 51.23 (1.70) | 14.61 (0.57) | 9.68 (0.38) | 3.61 (0.19) | 0.71 |
| | CoUn | 50.15 (0.62) | 14.67 (0.51) | 10.09 (0.03) | 3.28 (0.14) | 0.33 |

## E.2 Training and Unlearning Configurations

**Original Model $\theta_o$ Training.** Following [8, 9, 10], we train original models for all different datasets and model architectures for a total of 182 epochs. The batch size is set to 256. An SGD optimizer is used with an initial learning rate of 0.1 and a multi-step learning rate scheduler that reduces the learning rate by a factor of 10 at 50% and 75% of the training epochs. Momentum is set to 0.9, and weight decay is set to $5 \times 10^{-4}$. The transformation distribution used for data augmentation is described in Appendix E.3. For VGG-16, we use a linear warmup phase for the first 75 epochs.

**Unlearned Model $\theta_u$ Training.** For unlearning, all methods are run for 50 epochs. We used the SGD optimizer with a learning rate tuned within the range of [0.01, 0.1] for each MU method. A cosine annealing learning rate scheduler is used with a minimum learning rate set to $1 \times 10^{-4}$. Momentum is set to 0.9, and weight decay is set to $5 \times 10^{-4}$. The transformation distribution used for data augmentation is described in Appendix E.3. Additional details for each MU method are listed as follows:

- **BadT:** The temperature is set to 1.
- **SSD:** The weight selection is tuned in the interval [0.1, 100], while the dampening constant is tuned in the interval [0.1, 5].

- **NegGrad+:** The $\beta$ hyperparameter is tuned in the interval [0.95, 0.9999].
- **SCRUB:** The temperature is set to 1. The number of maximization steps is tuned in the interval [1, 10]. Both $\gamma$ and $\alpha$ are tuned in the interval [0.1, 3].
- **CU:** The contrastive scaling constant is tuned within the interval [0.1, 2] and the temperature is tuned within the interval of (0, 0.2]. The constant $\omega$ is tuned within the interval [1, 5].
- **$\ell_1$-sparse:** The $\ell_1$ regularization parameter is tuned in the interval $[10^{-4}, 10^{-1}]$. $\ell_1$ regularization is applied for 4 epochs, except for sequential forgetting experiments, where it is applied for 2 epochs and then reapplied every 10 epochs for a total of 50 epochs.
- **SalUn:** The mask threshold is tuned within the interval of [0.1, 1.0].
- **NoT:** For all model architectures, the first CNN layer (index: 0) is negated. For ViT [40], the positional representations and the second patch representation layer are negated (indices: 0 and 4).
- **CoUn:** The scaling constant $\lambda$ is tuned within the interval [0.1, 6], and the CL temperature $\tau$ is tuned within the interval of (0, 0.3].

All experiments are implemented using the PyTorch platform [43] and run using NVIDIA Tesla V100 GPUs.

### E.3 Data Augmentation

In our experiments, the following operations are applied sequentially to augment images:

- **Random cropping (C):** Output image size of $32 \times 32$ for CIFAR-10/100 and $64 \times 64$ for TinyImageNet, with a padding of 4 on each image border.
- **Random horizontal flip (H):** Applied with a probability of 0.5.
- **Color normalization (N):** Applied using mean values (0.4914, 0.4822, 0.4465) and standard deviations (0.2023, 0.1994, 0.2010).

However, in some ablation experiments, the following operations are added between horizontal flip and color normalization to augment images:

- **Random color jitter (J):** Applied with a probability of 0.8. Brightness, contrast, saturation, and hue are set to 0.8, 0.8, 0.8, and 0.2, respectively.
- **Random grayscale (G):** Applied with a probability of 0.2.

## F   Further Results

### F.1   Random Forgetting: Forget Ratio 10% (Additional Baselines)

Table 7 presents comparisons of CoUn with additional baseline methods: **BadT** (AAAI, 2023), **SSD** (AAAI, 2024), **SCRUB** (NeurIPS, 2023), and **CU** [33] using random data forgetting with a 10% forget ratio. CoUn consistently achieves superior performance compared to all baselines. Since BadT, SSD, SCRUB, and CU do not perform better than other baselines, we did not include them under different settings. Further, BadT and SCRUB demand higher computational resources due to their reliance on two teacher models (Original and Random) to guide the unlearned model.

### F.2   Class-wise Forgetting

Table 8 presents the results for class-wise forgetting. All baselines, including FT, exhibit minimal performance gaps with the Retrain model, suggesting that class-wise forgetting is relatively easy and can be effectively addressed using just FT (i.e., through catastrophic forgetting). This observation aligns with the findings from [11], which shows that lower entanglement between retain and forget data simplifies the unlearning task, making class-wise forgetting easier than random forgetting. This trend is further illustrated by the similarity between the representations of FT and CoUn in Figure 3 and those of the Retrain model in Figure 1. Nevertheless, CoUn achieves competitive results across all baselines. In this paper, the majority of our experiments focus on the more challenging unlearning scenarios (i.e., random forgetting).

Table 7: **Performance comparison of `CoUn` to additional baseline methods with 10% random data removal**. The gap (Δ) and the (best) average gap between each method and the Retrain model are reported.

| Dataset & Model | Method | Accuracy (%) | | | Efficacy (%) | Avg. Gap ↓ | Comp. Cost (PFLOPs) ↓ |
|---|---|---|---|---|---|---|---|
| | | Retain (Δ ↓) | Unlearn (Δ ↓) | Test (Δ ↓) | MIA (Δ ↓) | | |
| CIFAR-10 ResNet-18 | Retrain | $100.00_{\pm\,0.00}\,(0.00)$ | $4.81_{\pm\,0.27}\,(0.00)$ | $94.67_{\pm\,0.24}\,(0.00)$ | $11.02_{\pm\,0.58}\,(0.00)$ | 0.00 | 27.37 |
| | FT | $99.99_{\pm\,0.00}\,(0.01)$ | $3.76_{\pm\,0.31}\,(1.05)$ | $94.70_{\pm\,0.14}\,(0.03)$ | $9.51_{\pm\,0.28}\,(1.51)$ | 0.65 | 6.32 |
| | BadT | $99.94_{\pm\,0.03}\,(0.06)$ | $0.07_{\pm\,0.05}\,(4.74)$ | $94.05_{\pm\,0.15}\,(0.62)$ | $10.10_{\pm\,2.35}\,(0.92)$ | 1.58 | 9.89 |
| | SSD | $100.00_{\pm\,0.00}\,(0.00)$ | $0.02_{\pm\,0.00}\,(4.79)$ | $94.80_{\pm\,0.00}\,(0.13)$ | $0.62_{\pm\,0.00}\,(10.40)$ | 3.83 | 0.06 |
| | NegGrad+ | $99.95_{\pm\,0.02}\,(0.05)$ | $4.82_{\pm\,0.24}\,(0.01)$ | $94.32_{\pm\,0.23}\,(0.35)$ | $9.09_{\pm\,0.30}\,(1.93)$ | 0.58 | 6.02 |
| | SCRUB | $99.97_{\pm\,0.01}\,(0.03)$ | $3.93_{\pm\,0.23}\,(0.88)$ | $94.61_{\pm\,0.17}\,(0.06)$ | $9.53_{\pm\,0.34}\,(1.49)$ | 0.62 | 8.93 |
| | CU | $99.32_{\pm\,0.06}\,(0.68)$ | $5.48_{\pm\,0.16}\,(0.67)$ | $94.18_{\pm\,0.22}\,(0.49)$ | $11.63_{\pm\,0.72}\,(0.61)$ | 0.61 | 3.36 |
| | $\ell_1$-sparse | $99.97_{\pm\,0.01}\,(0.03)$ | $5.40_{\pm\,0.40}\,(0.59)$ | $93.81_{\pm\,0.21}\,(0.86)$ | $10.97_{\pm\,0.35}\,(0.05)$ | 0.38 | 6.92 |
| | SalUn | $99.10_{\pm\,0.35}\,(0.90)$ | $4.31_{\pm\,0.42}\,(0.50)$ | $93.84_{\pm\,0.27}\,(0.83)$ | $11.15_{\pm\,2.04}\,(0.13)$ | 0.59 | 8.66 |
| | NoT | $99.99_{\pm\,0.00}\,(0.01)$ | $4.19_{\pm\,0.25}\,(0.62)$ | $94.65_{\pm\,0.24}\,(0.02)$ | $10.45_{\pm\,0.51}\,(0.57)$ | 0.30 | 7.52 |
| | CoUn | $99.99_{\pm\,0.00}\,(0.01)$ | $4.12_{\pm\,0.31}\,(0.69)$ | $94.57_{\pm\,0.24}\,(0.10)$ | $10.81_{\pm\,0.31}\,(0.21)$ | 0.25 | 8.02 |

Table 8: **Performance comparison of `CoUn` to the baseline methods with class-wise 'truck' samples removal**. The gap (Δ) and the (best) average gap between each method and the Retrain model are reported.

| Dataset & Model | Method | Accuracy (%) | | | Efficacy (%) | Avg. Gap ↓ | Comp. Cost (PFLOPs) ↓ |
|---|---|---|---|---|---|---|---|
| | | Retain (Δ ↓) | Unlearn (Δ ↓) | Test (Δ ↓) | MIA (Δ ↓) | | |
| CIFAR-10 ResNet-18 | Retrain | $100.00_{\pm\,0.00}\,(0.00)$ | $100.00_{\pm\,0.00}\,(0.00)$ | $95.14_{\pm\,0.18}\,(0.00)$ | $100.00_{\pm\,0.00}\,(0.00)$ | 0.00 | 27.37 |
| | FT | $99.97_{\pm\,0.01}\,(0.03)$ | $100.00_{\pm\,0.00}\,(0.00)$ | $94.99_{\pm\,0.19}\,(0.15)$ | $100.00_{\pm\,0.00}\,(0.00)$ | 0.04 | 6.02 |
| | NegGrad+ | $99.98_{\pm\,0.02}\,(0.02)$ | $100.00_{\pm\,0.00}\,(0.00)$ | $95.10_{\pm\,0.18}\,(0.04)$ | $100.00_{\pm\,0.00}\,(0.00)$ | 0.02 | 9.03 |
| | $\ell_1$-sparse | $100.00_{\pm\,0.00}\,(0.00)$ | $100.00_{\pm\,0.00}\,(0.00)$ | $95.10_{\pm\,0.14}\,(0.04)$ | $100.00_{\pm\,0.00}\,(0.00)$ | 0.01 | 4.51 |
| | SalUn | $100.00_{\pm\,0.00}\,(0.00)$ | $100.00_{\pm\,0.00}\,(0.00)$ | $95.11_{\pm\,0.11}\,(0.03)$ | $100.00_{\pm\,0.00}\,(0.00)$ | 0.01 | 8.66 |
| | NoT | $100.00_{\pm\,0.00}\,(0.00)$ | $100.00_{\pm\,0.00}\,(0.00)$ | $95.14_{\pm\,0.13}\,(0.00)$ | $100.00_{\pm\,0.00}\,(0.00)$ | 0.00 | 6.02 |
| | CoUn | $100.00_{\pm\,0.00}\,(0.00)$ | $100.00_{\pm\,0.00}\,(0.00)$ | $95.18_{\pm\,0.20}\,(0.04)$ | $100.00_{\pm\,0.00}\,(0.00)$ | 0.01 | 9.03 |

## F.3 Integration of `CoUn`'s CL Module with Baselines: Detailed Results

Tables 9 and 10 provide detailed results for integrating `CoUn`'s CL module with existing baselines for 10% and 50% random data forgetting, respectively. Results show that adding our CL module significantly empowers the performance of existing MU methods, as measured by the average gap.

## F.4 Effect of CL Transformation on Representation Space

Huang et al. [35] provides a theoretical framework for understanding the generalization ability of CL, emphasizing the role of data augmentation and representation properties. They show that stronger transformations[1] in CL yield more compact and well-separated clusters in the representation space. These insights are particularly relevant to our observations on the effect of transformation strength in the context of MU.

In MU, the goal of CL is not to enhance clustering, but rather to weaken the clustering of forget data. Looser clustering allows their representations to be pushed toward clusters of other retain representations that are semantically similar to the forget representations, even if those clusters are different from the original ones of the forget data. Therefore, strong augmentations reduce the likelihood of cluster overlap, even in the presence of false negative samples.

As illustrated in Figure 9, weaker transformations lead to more effective unlearning. This is because they result in less tightly clustered representations, allowing forget data to overlap with retain data from other classes, particularity in the absence of supervised signals for forget data. Figure 12 further shows that stronger CL transformations (e.g., $\mathcal{T}_{CL}$ = CHJGN) produce tighter clusters of forget data compared to weaker ones (e.g., $\mathcal{T}_{CL}$ = CHN), consistent with the findings of [35]. However, from an MU perspective, such tight clustering impedes unlearning. Hence, using simpler CL transformations generally enhances unlearning effectiveness. This inverse relationship between representation compactness and unlearning efficacy is reinforced by the results in Figure 9.

---

[1]Transformations drawn from a distribution combining multiple data augmentations.

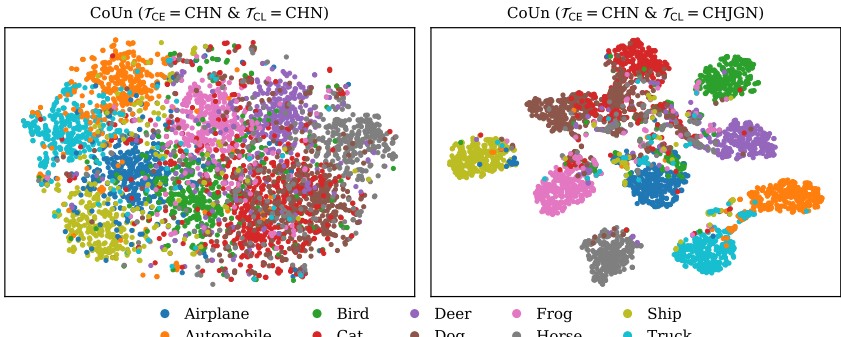

CoUn ($\mathcal{T}_{\text{CE}}$ = CHN & $\mathcal{T}_{\text{CL}}$ = CHN)     CoUn ($\mathcal{T}_{\text{CE}}$ = CHN & $\mathcal{T}_{\text{CL}}$ = CHJGN)

● Airplane   ● Bird   ● Deer   ● Frog   ● Ship
● Automobile   ● Cat   ● Dog   ● Horse   ● Truck

Figure 12: **Effect of CL transformation on forget data representations.** t-SNE visualizations of forget data representations extracted from the penultimate layer of $\theta_u$ on CIFAR-10 with ResNet-18 under a 50% forget data ratio. *Left*: CoUn with a simple CL transformation ($\mathcal{T}_{\text{CL}}$ = CHN). *Right*: CoUn with a strong CL transformation ($\mathcal{T}_{\text{CL}}$ = CHJGN). The transformation for supervised learning is fixed at $\mathcal{T}_{\text{CE}}$ = CHN. The CHJGN transformation applies a sequence of operations: crop, horizontal flip, color jitter, grayscale, and color normalization. Stronger transformations lead to tighter clustering of forget data representations, reducing unlearning effectiveness. Additionally, tighter clustering leads to less cluster overlap compared to weaker clustering (from simpler $\mathcal{T}_{\text{CL}}$).

Table 9: **Performance comparison when CoUn's contrastive learning (CL) module is integrated into baselines** under **10% random data forgetting.** The gap ($\Delta$) and the (best) average gap between each method and the Retrain model are reported.

| Dataset & Model | Method | CL | Accuracy (%) | | | Efficacy (%) | Avg. Gap ↓ | Comp. Cost (PFLOPs) ↓ |
|---|---|---|---|---|---|---|---|---|
| | | | Retain ($\Delta \downarrow$) | Unlearn ($\Delta \downarrow$) | Test ($\Delta \downarrow$) | MIA ($\Delta \downarrow$) | | |
| CIFAR-10 ResNet-18 | Retrain | - | $100.00_{\pm 0.00}(0.00)$ | $4.81_{\pm 0.27}(0.00)$ | $94.67_{\pm 0.24}(0.00)$ | $11.02_{\pm 0.58}(0.00)$ | 0.00 | 27.37 |
| | FT | ✗ | $99.99_{\pm 0.00}(0.01)$ | $3.76_{\pm 0.31}(1.05)$ | $94.70_{\pm 0.14}(0.03)$ | $9.51_{\pm 0.28}(1.51)$ | 0.65 | 6.32 |
| | | ✓ | $99.99_{\pm 0.00}(0.01)$ | $4.12_{\pm 0.31}(0.69)$ | $94.57_{\pm 0.24}(0.10)$ | $10.81_{\pm 0.31}(0.21)$ | 0.25 | 8.02 |
| | NegGrad+ | ✗ | $99.95_{\pm 0.02}(0.05)$ | $4.82_{\pm 0.24}(0.01)$ | $94.32_{\pm 0.23}(0.35)$ | $9.09_{\pm 0.30}(1.93)$ | 0.58 | 6.02 |
| | | ✓ | $99.98_{\pm 0.01}(0.02)$ | $5.21_{\pm 0.29}(0.40)$ | $94.55_{\pm 0.16}(0.12)$ | $10.30_{\pm 0.46}(0.72)$ | 0.32 | 7.52 |
| | $\ell_1$-sparse | ✗ | $99.97_{\pm 0.01}(0.03)$ | $5.40_{\pm 0.40}(0.59)$ | $93.81_{\pm 0.21}(0.86)$ | $10.97_{\pm 0.35}(0.05)$ | 0.38 | 6.92 |
| | | ✓ | $100.00_{\pm 0.00}(0.00)$ | $4.35_{\pm 0.19}(0.46)$ | $94.48_{\pm 0.18}(0.19)$ | $10.80_{\pm 0.40}(0.22)$ | 0.22 | 7.62 |
| | NoT | ✗ | $99.99_{\pm 0.00}(0.01)$ | $4.19_{\pm 0.25}(0.62)$ | $94.65_{\pm 0.24}(0.02)$ | $10.45_{\pm 0.51}(0.57)$ | 0.30 | 7.52 |
| | | ✓ | $100.00_{\pm 0.00}(0.00)$ | $4.20_{\pm 0.24}(0.61)$ | $94.65_{\pm 0.17}(0.02)$ | $11.08_{\pm 0.38}(0.06)$ | 0.17 | 7.52 |
| CIFAR-100 ResNet-18 | Retrain | - | $99.98_{\pm 0.00}(0.00)$ | $24.26_{\pm 0.53}(0.00)$ | $75.56_{\pm 0.26}(0.00)$ | $48.44_{\pm 0.36}(0.00)$ | 0.00 | 27.37 |
| | FT | ✗ | $99.97_{\pm 0.00}(0.01)$ | $16.39_{\pm 0.60}(7.87)$ | $76.75_{\pm 0.25}(1.19)$ | $44.06_{\pm 0.58}(4.38)$ | 3.36 | 7.22 |
| | | ✓ | $99.97_{\pm 0.00}(0.01)$ | $22.01_{\pm 0.44}(2.25)$ | $72.88_{\pm 0.39}(2.68)$ | $47.82_{\pm 0.96}(0.62)$ | 1.39 | 9.63 |
| | NegGrad+ | ✗ | $99.96_{\pm 0.01}(0.02)$ | $30.09_{\pm 0.41}(5.83)$ | $75.46_{\pm 0.36}(0.10)$ | $47.72_{\pm 0.32}(0.72)$ | 1.67 | 7.62 |
| | | ✓ | $99.98_{\pm 0.00}(0.00)$ | $22.48_{\pm 0.56}(1.78)$ | $74.52_{\pm 0.26}(1.04)$ | $48.32_{\pm 0.54}(0.12)$ | 0.74 | 12.53 |
| | $\ell_1$-sparse | ✗ | $99.95_{\pm 0.01}(0.03)$ | $23.94_{\pm 0.50}(0.32)$ | $74.95_{\pm 0.32}(0.61)$ | $42.81_{\pm 0.56}(5.63)$ | 1.65 | 7.22 |
| | | ✓ | $99.96_{\pm 0.01}(0.02)$ | $24.57_{\pm 0.38}(0.31)$ | $74.46_{\pm 0.20}(1.10)$ | $44.62_{\pm 0.61}(3.82)$ | 1.31 | 9.63 |
| | NoT | ✗ | $99.97_{\pm 0.01}(0.01)$ | $17.99_{\pm 0.40}(6.27)$ | $76.27_{\pm 0.24}(0.71)$ | $44.28_{\pm 0.57}(4.16)$ | 2.79 | 7.22 |
| | | ✓ | $99.97_{\pm 0.00}(0.01)$ | $21.53_{\pm 0.53}(2.73)$ | $73.61_{\pm 0.39}(1.95)$ | $46.84_{\pm 0.80}(1.60)$ | 1.57 | 9.63 |
| TinyImageNet ResNet-18 | Retrain | - | $99.98_{\pm 0.00}(0.00)$ | $36.16_{\pm 0.35}(0.00)$ | $63.82_{\pm 0.20}(0.00)$ | $63.73_{\pm 0.42}(0.00)$ | 0.00 | 218.98 |
| | FT | ✗ | $99.98_{\pm 0.00}(0.00)$ | $32.76_{\pm 0.42}(3.40)$ | $64.65_{\pm 0.29}(0.83)$ | $56.93_{\pm 0.59}(6.80)$ | 2.76 | 60.16 |
| | | ✓ | $99.95_{\pm 0.01}(0.03)$ | $35.10_{\pm 0.30}(1.06)$ | $63.27_{\pm 0.12}(0.55)$ | $57.57_{\pm 0.17}(6.16)$ | 1.95 | 80.21 |
| | NegGrad+ | ✗ | $99.98_{\pm 0.00}(0.00)$ | $38.01_{\pm 0.32}(1.85)$ | $64.68_{\pm 0.26}(0.86)$ | $57.84_{\pm 0.47}(5.89)$ | 2.15 | 80.21 |
| | | ✓ | $99.96_{\pm 0.01}(0.02)$ | $36.68_{\pm 0.13}(0.52)$ | $63.73_{\pm 0.17}(0.09)$ | $57.87_{\pm 0.36}(5.86)$ | 1.62 | 100.27 |
| | $\ell_1$-sparse | ✗ | $99.96_{\pm 0.00}(0.02)$ | $36.96_{\pm 0.37}(0.80)$ | $62.62_{\pm 0.39}(1.20)$ | $56.74_{\pm 0.46}(6.99)$ | 2.25 | 60.16 |
| | | ✓ | $99.97_{\pm 0.00}(0.01)$ | $36.33_{\pm 0.41}(0.17)$ | $63.27_{\pm 0.32}(0.55)$ | $56.96_{\pm 0.33}(6.77)$ | 1.88 | 80.21 |
| | NoT | ✗ | $99.98_{\pm 0.00}(0.00)$ | $35.64_{\pm 0.71}(0.52)$ | $63.66_{\pm 0.70}(0.16)$ | $56.08_{\pm 0.93}(7.65)$ | 2.08 | 80.21 |
| | | ✓ | $99.94_{\pm 0.01}(0.04)$ | $36.19_{\pm 0.47}(0.03)$ | $63.06_{\pm 0.33}(0.76)$ | $56.70_{\pm 0.41}(7.03)$ | 1.97 | 80.21 |
| CIFAR-100 VGG-16 | Retrain | - | $99.75_{\pm 0.07}(0.00)$ | $33.23_{\pm 0.38}(0.00)$ | $67.07_{\pm 0.57}(0.00)$ | $40.69_{\pm 0.40}(0.00)$ | 0.00 | 15.58 |
| | FT | ✗ | $99.26_{\pm 0.05}(0.49)$ | $26.02_{\pm 0.55}(7.21)$ | $68.42_{\pm 0.32}(1.35)$ | $35.51_{\pm 0.62}(5.18)$ | 3.56 | 3.42 |
| | | ✓ | $99.82_{\pm 0.01}(0.07)$ | $32.37_{\pm 0.46}(0.86)$ | $63.80_{\pm 0.35}(3.27)$ | $39.64_{\pm 0.25}(1.05)$ | 1.31 | 5.71 |
| | NegGrad+ | ✗ | $94.92_{\pm 0.41}(4.83)$ | $35.44_{\pm 0.62}(2.21)$ | $65.54_{\pm 0.39}(1.53)$ | $40.67_{\pm 0.60}(0.02)$ | 2.15 | 3.42 |
| | | ✓ | $98.43_{\pm 0.44}(1.32)$ | $34.65_{\pm 0.91}(1.42)$ | $65.98_{\pm 0.51}(1.09)$ | $39.12_{\pm 1.29}(1.57)$ | 1.35 | 5.71 |
| | $\ell_1$-sparse | ✗ | $99.27_{\pm 0.04}(0.48)$ | $26.96_{\pm 0.66}(6.27)$ | $68.01_{\pm 0.37}(0.94)$ | $35.31_{\pm 0.50}(5.38)$ | 3.27 | 3.42 |
| | | ✓ | $98.94_{\pm 0.05}(0.81)$ | $29.70_{\pm 0.55}(3.53)$ | $66.12_{\pm 0.28}(0.95)$ | $37.28_{\pm 0.60}(3.41)$ | 2.17 | 5.71 |
| | NoT | ✗ | $96.17_{\pm 4.28}(3.58)$ | $30.11_{\pm 3.02}(3.12)$ | $66.75_{\pm 1.73}(0.32)$ | $36.47_{\pm 1.18}(4.22)$ | 2.81 | 4.28 |
| | | ✓ | $98.72_{\pm 0.43}(1.03)$ | $35.25_{\pm 1.96}(2.02)$ | $61.90_{\pm 1.39}(5.17)$ | $40.51_{\pm 0.94}(0.18)$ | 2.10 | 4.57 |
| CIFAR-100 ViT | Retrain | - | $99.97_{\pm 0.00}(0.00)$ | $38.73_{\pm 0.69}(0.00)$ | $61.89_{\pm 0.62}(0.00)$ | $61.75_{\pm 0.33}(0.00)$ | 0.00 | 86.83 |
| | FT | ✗ | $99.78_{\pm 0.04}(0.19)$ | $10.83_{\pm 0.41}(27.90)$ | $61.12_{\pm 0.45}(0.77)$ | $31.50_{\pm 0.42}(30.25)$ | 14.78 | 5.79 |
| | | ✓ | $99.91_{\pm 0.03}(0.06)$ | $36.81_{\pm 1.08}(1.92)$ | $56.49_{\pm 0.55}(5.40)$ | $53.92_{\pm 0.42}(7.83)$ | 3.80 | 19.29 |
| | NegGrad+ | ✗ | $99.88_{\pm 0.03}(0.09)$ | $45.26_{\pm 0.41}(6.53)$ | $59.33_{\pm 0.64}(2.56)$ | $55.00_{\pm 0.40}(6.75)$ | 3.98 | 11.58 |
| | | ✓ | $99.96_{\pm 0.01}(0.01)$ | $38.71_{\pm 0.46}(0.02)$ | $59.07_{\pm 0.31}(2.82)$ | $52.38_{\pm 0.43}(9.37)$ | 3.05 | 24.12 |
| | $\ell_1$-sparse | ✗ | $99.32_{\pm 0.04}(0.65)$ | $31.71_{\pm 0.52}(7.02)$ | $63.33_{\pm 0.32}(1.44)$ | $46.49_{\pm 0.82}(15.26)$ | 6.09 | 14.47 |
| | | ✓ | $99.63_{\pm 0.03}(0.34)$ | $39.07_{\pm 0.53}(0.34)$ | $58.18_{\pm 0.39}(3.71)$ | $50.26_{\pm 0.24}(11.49)$ | 3.97 | 19.29 |
| | NoT | ✗ | $99.89_{\pm 0.02}(0.08)$ | $20.29_{\pm 1.93}(18.44)$ | $61.82_{\pm 0.29}(0.07)$ | $43.55_{\pm 1.36}(18.20)$ | 9.20 | 8.68 |
| | | ✓ | $99.93_{\pm 0.01}(0.04)$ | $37.90_{\pm 0.72}(0.83)$ | $58.85_{\pm 0.35}(3.04)$ | $56.50_{\pm 0.77}(5.25)$ | 2.29 | 19.29 |

Table 10: **Performance comparison when `CoUn`'s contrastive learning (CL) module is integrated into baselines** under **50% random data forgetting.** The gap (Δ) and the (best) average gap between each method and the Retrain model are reported.

| Dataset & Model | Method | CL | Accuracy (%) | | | Efficacy (%) | Avg. Gap ↓ | Comp. Cost (PFLOPs) ↓ |
|---|---|---|---|---|---|---|---|---|
| | | | Retain (Δ↓) | Unlearn (Δ↓) | Test (Δ↓) | MIA (Δ↓) | | |
| CIFAR-10 ResNet-18 | Retrain | - | $100.00_{\pm 0.00}(0.00)$ | $7.29_{\pm 0.36}(0.00)$ | $92.28_{\pm 0.23}(0.00)$ | $17.33_{\pm 0.65}(0.00)$ | 0.00 | 15.24 |
| | FT | ✗ | $99.38_{\pm 0.24}(0.62)$ | $6.32_{\pm 0.41}(0.97)$ | $91.91_{\pm 0.41}(0.37)$ | $12.64_{\pm 0.51}(4.69)$ | 1.66 | 2.51 |
| | | ✓ | $99.97_{\pm 0.03}(0.03)$ | $6.19_{\pm 0.30}(1.10)$ | $92.36_{\pm 0.26}(0.08)$ | $16.94_{\pm 0.48}(0.39)$ | 0.40 | 3.35 |
| | NegGrad+ | ✗ | $100.00_{\pm 0.00}(0.00)$ | $4.06_{\pm 0.20}(0.75)$ | $93.81_{\pm 0.23}(0.86)$ | $9.05_{\pm 0.22}(1.97)$ | 0.90 | 5.58 |
| | | ✓ | $100.00_{\pm 0.00}(0.00)$ | $4.65_{\pm 0.25}(0.16)$ | $93.45_{\pm 0.24}(1.22)$ | $11.45_{\pm 0.35}(0.43)$ | 0.45 | 5.58 |
| | $\ell_1$-sparse | ✗ | $99.77_{\pm 0.03}(0.23)$ | $9.02_{\pm 0.16}(1.73)$ | $90.66_{\pm 0.24}(1.62)$ | $16.05_{\pm 0.29}(1.28)$ | 1.22 | 4.19 |
| | | ✓ | $100.00_{\pm 0.00}(0.00)$ | $6.91_{\pm 0.23}(0.38)$ | $92.30_{\pm 0.19}(0.02)$ | $17.14_{\pm 0.52}(0.19)$ | 0.15 | 4.47 |
| | NoT | ✗ | $99.98_{\pm 0.01}(0.02)$ | $5.95_{\pm 0.18}(1.34)$ | $92.84_{\pm 0.18}(0.56)$ | $13.90_{\pm 0.37}(3.43)$ | 1.34 | 3.35 |
| | | ✓ | $99.99_{\pm 0.01}(0.01)$ | $6.84_{\pm 1.21}(0.45)$ | $91.64_{\pm 0.73}(0.64)$ | $17.00_{\pm 1.46}(0.33)$ | 0.36 | 5.58 |
| CIFAR-100 ResNet-18 | Retrain | - | $99.98_{\pm 0.01}(0.00)$ | $31.41_{\pm 0.40}(0.00)$ | $68.41_{\pm 0.34}(0.00)$ | $58.35_{\pm 0.53}(0.00)$ | 0.00 | 15.24 |
| | FT | ✗ | $99.98_{\pm 0.01}(0.00)$ | $17.36_{\pm 0.19}(14.05)$ | $74.16_{\pm 0.39}(5.75)$ | $50.60_{\pm 0.42}(7.75)$ | 6.89 | 4.19 |
| | | ✓ | $99.98_{\pm 0.01}(0.00)$ | $31.43_{\pm 1.75}(0.02)$ | $65.60_{\pm 0.71}(2.81)$ | $55.99_{\pm 1.18}(2.36)$ | 1.30 | 5.58 |
| | NegGrad+ | ✗ | $99.98_{\pm 0.01}(0.00)$ | $26.32_{\pm 0.21}(5.09)$ | $71.98_{\pm 0.30}(3.57)$ | $52.32_{\pm 0.36}(6.03)$ | 3.67 | 5.36 |
| | | ✓ | $99.98_{\pm 0.01}(0.00)$ | $31.55_{\pm 0.39}(0.14)$ | $68.34_{\pm 0.30}(0.07)$ | $56.34_{\pm 0.45}(2.01)$ | 0.55 | 6.70 |
| | $\ell_1$-sparse | ✗ | $99.94_{\pm 0.02}(0.04)$ | $32.26_{\pm 0.23}(0.85)$ | $67.66_{\pm 0.35}(0.75)$ | $51.54_{\pm 0.29}(6.81)$ | 2.11 | 4.19 |
| | | ✓ | $99.98_{\pm 0.00}(0.00)$ | $29.93_{\pm 0.27}(1.48)$ | $68.35_{\pm 0.39}(0.06)$ | $54.84_{\pm 0.55}(3.51)$ | 1.26 | 5.58 |
| | NoT | ✗ | $98.64_{\pm 0.43}(1.34)$ | $26.43_{\pm 0.75}(4.98)$ | $67.97_{\pm 0.83}(0.44)$ | $43.82_{\pm 0.60}(14.53)$ | 5.32 | 2.01 |
| | | ✓ | $99.98_{\pm 0.01}(0.00)$ | $26.01_{\pm 0.42}(5.40)$ | $69.01_{\pm 0.46}(0.60)$ | $54.30_{\pm 0.53}(4.05)$ | 2.51 | 5.58 |
| TinyImageNet ResNet-18 | Retrain | - | $99.99_{\pm 0.00}(0.00)$ | $43.01_{\pm 0.20}(0.00)$ | $57.28_{\pm 0.43}(0.00)$ | $71.22_{\pm 0.17}(0.00)$ | 0.00 | 121.93 |
| | FT | ✗ | $99.99_{\pm 0.00}(0.00)$ | $36.78_{\pm 0.18}(6.23)$ | $60.59_{\pm 0.38}(3.31)$ | $66.28_{\pm 0.20}(4.94)$ | 3.62 | 33.50 |
| | | ✓ | $99.98_{\pm 0.01}(0.01)$ | $43.21_{\pm 1.57}(0.20)$ | $55.75_{\pm 1.34}(1.53)$ | $66.59_{\pm 0.41}(4.63)$ | 1.59 | 44.66 |
| | NegGrad+ | ✗ | $99.99_{\pm 0.00}(0.00)$ | $47.62_{\pm 0.25}(4.61)$ | $58.85_{\pm 0.32}(1.57)$ | $66.43_{\pm 0.33}(4.79)$ | 2.74 | 33.50 |
| | | ✓ | $99.98_{\pm 0.01}(0.01)$ | $45.78_{\pm 0.19}(2.77)$ | $56.68_{\pm 0.36}(0.60)$ | $66.77_{\pm 0.28}(4.45)$ | 1.96 | 55.83 |
| | $\ell_1$-sparse | ✗ | $99.99_{\pm 0.00}(0.00)$ | $38.83_{\pm 0.21}(4.18)$ | $60.25_{\pm 0.30}(2.97)$ | $65.82_{\pm 0.21}(5.40)$ | 3.14 | 33.50 |
| | | ✓ | $99.94_{\pm 0.01}(0.05)$ | $42.11_{\pm 0.22}(0.90)$ | $57.02_{\pm 0.43}(0.26)$ | $65.92_{\pm 0.26}(5.30)$ | 1.63 | 44.66 |
| | NoT | ✗ | $99.99_{\pm 0.00}(0.00)$ | $40.94_{\pm 0.43}(2.07)$ | $58.27_{\pm 0.39}(0.99)$ | $66.23_{\pm 0.36}(4.99)$ | 2.01 | 33.50 |
| | | ✓ | $99.99_{\pm 0.00}(0.00)$ | $42.11_{\pm 0.29}(0.90)$ | $57.15_{\pm 0.51}(0.13)$ | $65.91_{\pm 0.19}(5.31)$ | 1.58 | 44.66 |
| CIFAR-100 VGG-16 | Retrain | - | $99.65_{\pm 0.18}(0.00)$ | $42.85_{\pm 0.54}(0.00)$ | $57.70_{\pm 0.47}(0.00)$ | $50.19_{\pm 0.93}(0.00)$ | 0.00 | 8.67 |
| | FT | ✗ | $97.71_{\pm 0.25}(1.94)$ | $29.82_{\pm 0.58}(13.03)$ | $63.72_{\pm 0.43}(6.02)$ | $39.98_{\pm 0.62}(10.21)$ | 7.80 | 1.43 |
| | | ✓ | $99.88_{\pm 0.02}(0.23)$ | $42.37_{\pm 0.80}(0.48)$ | $55.19_{\pm 0.68}(2.51)$ | $50.00_{\pm 0.68}(0.19)$ | 0.85 | 3.18 |
| | NegGrad+ | ✗ | $95.54_{\pm 0.56}(4.11)$ | $43.42_{\pm 0.38}(0.57)$ | $58.52_{\pm 0.44}(0.82)$ | $43.51_{\pm 0.36}(6.68)$ | 3.04 | 3.18 |
| | | ✓ | $96.70_{\pm 0.00}(2.95)$ | $42.92_{\pm 0.00}(0.07)$ | $59.14_{\pm 0.00}(1.44)$ | $46.61_{\pm 0.00}(3.58)$ | 2.01 | 3.18 |
| | $\ell_1$-sparse | ✗ | $98.25_{\pm 1.53}(1.40)$ | $34.24_{\pm 1.87}(8.61)$ | $62.76_{\pm 1.65}(5.06)$ | $42.12_{\pm 0.55}(8.07)$ | 5.79 | 1.91 |
| | | ✓ | $99.34_{\pm 0.23}(0.31)$ | $42.78_{\pm 0.38}(0.07)$ | $55.06_{\pm 0.39}(2.64)$ | $46.41_{\pm 0.50}(3.78)$ | 1.70 | 3.18 |
| | NoT | ✗ | $94.23_{\pm 7.94}(5.42)$ | $34.64_{\pm 7.05}(8.21)$ | $61.58_{\pm 4.67}(3.88)$ | $39.84_{\pm 1.62}(10.35)$ | 6.96 | 2.38 |
| | | ✓ | $99.55_{\pm 0.19}(0.10)$ | $42.78_{\pm 2.71}(0.07)$ | $55.17_{\pm 1.71}(2.53)$ | $48.88_{\pm 1.92}(1.31)$ | 1.00 | 3.18 |
| CIFAR-100 ViT | Retrain | - | $99.98_{\pm 0.01}(0.00)$ | $48.07_{\pm 0.33}(0.00)$ | $52.40_{\pm 0.58}(0.00)$ | $69.54_{\pm 0.29}(0.00)$ | 0.00 | 48.35 |
| | FT | ✗ | $98.71_{\pm 0.30}(1.27)$ | $10.91_{\pm 0.96}(37.16)$ | $56.79_{\pm 0.73}(4.39)$ | $28.18_{\pm 0.93}(41.36)$ | 21.05 | 1.61 |
| | | ✓ | $99.71_{\pm 0.62}(0.27)$ | $45.95_{\pm 3.70}(2.12)$ | $49.45_{\pm 2.15}(2.95)$ | $59.24_{\pm 1.48}(10.30)$ | 3.91 | 10.74 |
| | NegGrad+ | ✗ | $99.30_{\pm 0.13}(0.68)$ | $45.35_{\pm 0.48}(2.72)$ | $50.82_{\pm 0.47}(1.58)$ | $55.07_{\pm 0.33}(14.47)$ | 4.86 | 6.45 |
| | | ✓ | $99.14_{\pm 0.13}(0.84)$ | $45.78_{\pm 0.38}(2.29)$ | $50.93_{\pm 0.39}(1.47)$ | $57.96_{\pm 0.50}(11.57)$ | 4.04 | 5.37 |
| | $\ell_1$-sparse | ✗ | $71.18_{\pm 0.57}(28.80)$ | $47.30_{\pm 0.26}(0.77)$ | $53.32_{\pm 0.52}(0.92)$ | $44.22_{\pm 2.98}(25.32)$ | 13.95 | 8.06 |
| | | ✓ | $95.83_{\pm 0.68}(4.15)$ | $49.36_{\pm 0.19}(1.29)$ | $50.87_{\pm 0.54}(1.53)$ | $53.66_{\pm 0.75}(15.88)$ | 5.71 | 10.74 |
| | NoT | ✗ | $97.86_{\pm 1.71}(2.12)$ | $31.81_{\pm 2.05}(16.26)$ | $55.51_{\pm 1.15}(3.11)$ | $48.85_{\pm 2.48}(20.69)$ | 10.55 | 3.22 |
| | | ✓ | $99.93_{\pm 0.02}(0.05)$ | $50.21_{\pm 0.66}(2.14)$ | $49.50_{\pm 0.69}(2.90)$ | $65.80_{\pm 0.44}(3.74)$ | 2.21 | 10.74 |

# G  Broader Impacts

Research in machine unlearning holds significant societal value by empowering users to request the removal of their data from models and enhancing model safety and fairness through the elimination of harmful or outdated information. This work is exploratory in nature—we propose an approximate unlearning framework that uses contrastive learning to push forget representations into clusters of other retain samples that are semantically similar to the forget samples. Due to cluster collision, these retain samples may belong to clusters different from the original clusters of the forget samples. Given the nature of our approach, we do not foresee any direct negative societal impacts stemming from this work.

# H  Limitations and Future Work

While `CoUn` marks a step forward in leveraging CL for unlearning, its evaluation is currently limited to vision-based classification tasks on relatively small datasets (CIFAR-10/100 [36] and TinyImageNet [37]). Future work could explore its scalability to larger datasets such as ImageNet [44] and its applicability to other domains, including natural language processing. Additionally, although `CoUn` uses InfoNCE as the contrastive loss, investigating alternative self-supervised objectives, such as cross-correlation-based losses [31, 32], on unlearning performance presents an interesting direction

for future research. Moreover, like prior methods [8, 9, 10], `CoUn` relies heavily on access to retain data for effective unlearning. Investigating performance of approximate unlearning methods under partial access to retain data would be an important direction for future research. Finally, while CL introduces additional computational overhead due to augmented data and extra forward passes, `CoUn` consistently outperforms baselines even when computational budgets are matched. Future work could explore strategies to reduce this cost without sacrificing performance.

