# OpenReview forum: "CoUn: Empowering Machine Unlearning via Contrastive Learning"
_NeurIPS.cc/2025/Conference — NeurIPS 2025 poster_

### Official Review · Reviewer_6v6H · 2025-06-30

**Clarity:** 2
**Significance:** 2
**Originality:** 2
**Rating:** 4
**Confidence:** 4

**Summary:**

This paper introduces a novel approximate unlearning method leveraging the advantage of contrastive learning to manipulate the similarity of retain and forget data. The paper challenged the traditional contrastive learning can indirectly push the forget data to the clusters of retain data and proposed additional supervised learning loss function to cluster the forget data. Therefore, the contrastive learning is pushing away the entire cluster of forget from the retain. This paper also implements extensive experiments and intuitive visualization to prove the effectiveness of proposed method

**Questions:**

1.	The theoretical analysis is hard to follow. Notations are missing, (e.g. R[\epsilon] ) I guess this is the probability that the encoder representation to generate two views differences larger than $\epsilon$, which is a typical Lipchitz equation definition. But there is no exact notation for that.
2.	What is $\mu_{l}^{T}$?  and What is $\rho^{max}$. Althought the author mention that the proof of Theorem 1 can be found at paper [34], I would still suggest to place it under appendix and show clearly how (e.g. R[\epsilon]) interact with Theorem 1.
3.	In line 648, I am not sure how author ensures the equation (10)  $L_r  \leq L_u \leq L$ holds in general. More clarifications needed.
Experiments in Table 2 can apparently be extended to more data drops (e.g. 20%, 30%, 40%, 50%...). 10% data forgetting may not be representative enough. It will be interesting to see comparison on class-wise forgetting with comprehensive baselines. Table 1 only implement on CIFAR-10 with FT and Retrain baseline.

**Ethical Concerns:**

["NO or VERY MINOR ethics concerns only"]

**Limitations:**

yes

**Quality:**

2

**Strengths And Weaknesses:**

Strengths
The motivation of the work is clear. The results in Table 1 and Figure 4 look promising demonstrating effectiveness of the work.  Introduction, Technical Method and Experiment are easy to follow. The method is simple, well-motivated and efficient enough. More importantly, the proposed method can be integrated with other baseline for better unlearning performance.

Weaknesses
Theoretical contribution is weak as the main part of the theory came from other existing paper and the author did not state and explain the theory clearly.
1.	The theoretical analysis is hard to follow. Notations are missing, (e.g. R[\epsilon] ) I guess this is the probability that the encoder representation to generate two views differences larger than $\epsilon$, which is a typical Lipchitz equation definition. But there is no exact notation for that.
2.	What is $\mu_{l}^{T}$?  and What is $\rho^{max}$. Althought the author mention that the proof of Theorem 1 can be found at paper [34], I would still suggest to place it under appendix and show clearly how (e.g. R[\epsilon]) interact with Theorem 1.
3.	In line 648, I am not sure how author ensures the equation (10)  $L_r  \leq L_u \leq L$ holds in general. More clarifications needed.
Experiments in Table 2 can apparently be extended to more data drops (e.g. 20%, 30%, 40%, 50%...). 10% data forgetting may not be representative enough. It will be interesting to see comparison on class-wise forgetting with comprehensive baselines. Table 1 only implement on CIFAR-10 with FT and Retrain baseline.

---

> ### Author Rebuttal · Authors · 2025-07-30
>
> We thank the reviewer for their constructive comments and for noting that our work is **novel**, **challenging**, **simple**, **well-motivated**, implements **extensive experiments**, is **efficient**, and provides **intuitive visualization demonstrating effectiveness**. We appreciate the acknowledgment that the **introduction, technical method and experiment are easy to follow** and that our approach **can be integrated with other baselines for enhanced unlearning performance**. Below, we address their specific questions.
>
> $
> \textcolor{white}{f^{*}}
> $
>
> **Authors' response to Q1:** We appreciate the reviewer’s observation. While our intention was to define all notations, including $R[\epsilon]$, we acknowledge that defining them solely through mathematical expressions may not have been sufficiently explicit. To improve clarity and facilitate understanding of the theoretical analysis, we will revise the manuscript to clearly and formally define all variables at their first occurrence. Specifically, as shown in Eq. (5), $R[\epsilon]$ denotes the probability that, when a sample is drawn from the dataset, the distance between the encoder representations of its two augmented views exceeds $\epsilon$. Thus, a small value of $R[\epsilon]$ indicates good alignment in the representation space.
>
> $
> \textcolor{white}{f^{*}}
> $
>
> **Authors' response to Q2:** We thank the reviewer for pointing this out.
>
> 1. $\mu_l$ denotes the center of class $C_l$. It is defined in Line 240, and $\mu_l^T$ refers to its transpose.
>
> 2. $\rho^{\text{max}}$ is defined in Line 245 as a function of $(\sigma, \delta)$, which characterize the data augmentation, as well as the threshold $\epsilon$. Intuitively, better alignment (i.e., smaller $R[\epsilon]$) and a sharper concentration of augmented samples (i.e., larger $\sigma$ for a given $\delta$) result in a lower value of $\rho^{\text{max}}$.
>
> 3. Following the reviewer’s suggestion, we will include the full proof of Theorem 1 in the appendix and explicitly clarify how quantities such as $R[\epsilon]$, $\rho^{\text{max}}$, and class means interact in the derivation. This will enhance the clarity and self-containment of the theoretical analysis.
>
> $
> \textcolor{white}{f^{*}}
> $
>
> **Authors' response to Q3.1:** In Line 243, we state that the feature extractor $f_{\theta_u}$ is an $L$-Lipschitz function. That is, for any input sample (whether from the retain or forget data), the inequality in Eq. (8) holds. This implies that $f_{\theta_u}$ is globally Lipschitz with constant $L$, and therefore the Lipschitz constant over the forget data (i.e., $L_u$) must satisfy $L_u \leq L$.
>
> Moreover, as noted in Lines 644-648, training $f_{\theta_u}$ exclusively on the retained data encourages the function to vary more smoothly on those data. This indicates that $f_{\theta_u}$ exhibits greater stability and lower sensitivity on the retained data compared to the forgotten data. This leads to the inequality $L_r \leq L_u$.
>
> Taken together, these observations support the bound $L_r \leq L_u \leq L$.
>
> $
> \textcolor{white}{f^{*}}
> $
>
> **Authors' response to Q3.2:** Regarding the reviewers suggestion about having experiments with "more data drops", we would like to refer the reviewer to Table 3 in our paper, where we repeated all experiments in Table 2 but with a 50% forget data ratio. Additionally, Figures 4 and 5 also present results for both 10% (left) and 50% (right) forget ratios. Moreover, Figure 6 demonstrates a sequential data removal, where 10% of the data is removed initially, followed by an additional 10% every 10 epochs, reaching up to 50% (i.e., 10\% $\rightarrow$ 20\% $\rightarrow$ 30\% $\rightarrow$ 40\% $\rightarrow$ 50\%).
>
> $
> \textcolor{white}{f^{*}}
> $
>
> **Authors' response to Q3.3:** Regarding experiments related to **class-wise forgetting**, we would like to refer the reviewer to Section E.2, where we discussed the class-wise forgetting. In Table 5, we compare our method, CoUn, with state-of-the-art baselines under class-wise forgetting. Results demonstrate CoUn's competitive performance under class-wise forgetting.
>
> $
> \textcolor{white}{f^{*}}
> $
>
> **Authors' response to Q3.4:** We thank the reviewer for pointing the following out: "Table 1 only implement on CIFAR-10 with FT and Retrain baseline".
>
> Following the reviewer’s suggestion  and to further validate the robustness of our findings, we conducted additional experiments to include the missing baselines and we also conducted new experiments with another model architecture and dataset. These comprehensive results are presented in the Table below. These extended results will be reflected in the paper. As expected, CoUn yields predictions more aligned with the Retrain model compared to other baselines, further demonstrating that it better mimics the Retrain model in terms of classifying the forget samples based on semantic similarity.
>
> **Table: Predictions of forget 'truck' samples based on most semantically similar classes. The difference ($\Delta$) and the (best) average difference between each method and Retrain are reported.**
> |Forgetting Scenario|Method|Truck|Automobile|Airplane|Ship|Avg. Diff. ↓|
> |-|-|-|-|-|-|-|
> |Random (10%), CIFAR-10, ResNet-18|Original|100.00|0.00|0.00|0.00|-|
> ||Retrain|97.42 (0.00)|1.23 (0.00)|0.38 (0.00)|0.40 (0.00)|0.00|
> ||FT|98.12 (0.70)|0.75 (0.48)|0.36 (0.02)|0.32 (0.08)|0.32|
> ||NegGrad+|97.86 (0.44)|0.97 (0.26)|0.42 (0.04)|0.30 (0.10)|0.21|
> ||$\ell_1$-sparse|98.06 (0.64)|0.85 (0.38)|0.30 (0.08)| 0.38 (0.02)|0.28|
> ||SaUn|96.76 (0.66)|0.88 (0.35)| 0.34 (0.04)|0.34 (0.06)| 0.28|
> ||NoT|97.88 (0.56)|0.89 (0.34)| 0.40 (0.02)| 0.32 (0.08)| 0.25|
> ||**CoUn (Ours)**|97.84 (0.42)| 0.99 (0.24)| 0.32 (0.06)|0.42 (0.02)|**0.19**|
> ||
> |Random (10%), CIFAR-100, VGG-16|Original| 100.00|0.00|0.00|0.00|-|
> ||Retrain|49.53 (0.00)|15.18 (0.00)|10.06 (0.00)|3.42 (0.00)|0.00|
> ||FT|60.53 (11.00)|9.11 (6.07)|9.68 (0.38) |3.61 (0.19)|4.41|
> ||NegGrad+|41.94 (7.59)|13.47 (1.71)|12.71 (2.65)|4.74 (1.32)|3.32|
> ||$\ell_1$-sparse|57.69 (8.16)|13.47 (1.71)|10.06 (0.00)|2.85 (0.57)|2.61|
> ||SaUn|55.16 (5.63)|13.52 (1.66)| 9.68 (0.38)|2.63 (0.79)|2.12|
> ||NoT|51.23 (1.70)|14.61 (0.57)|9.68 (0.38)|3.61 (0.19)|0.71|
> ||**CoUn (Ours)**|50.15 (0.62)|14.67 (0.51)|10.09 (0.03)|3.28 (0.14)|**0.33**|

---

### Official Review · Reviewer_1SXk · 2025-06-30

**Clarity:** 3
**Significance:** 3
**Originality:** 2
**Rating:** 4
**Confidence:** 3

**Summary:**

This paper proposed applying the contrastive learning method assisted with supervised learning method exclusively on retain data to conduct machine unlearning tasks. The experimental results demonstrated that the proposed approach achieved superior performance equivalence w.r.t. the gold-standard retrain model.

**Questions:**

1. It would be good to conduct experiments to understand how likely CL would suffer from excessive or insufficient perturbation, for example, with longer or less training epochs.
2. How sensitive is the $\lambda$ choice? Are the reported results in Table 2 using the same $\lambda$? If not, how much performance loss would be introduced if using the same $\lambda$ (from a careless hyperparameter tuning).
3. Does the reported computation cost consider the hyperparameter tuning costs for both baseline methods and proposed methods?
4. What would happen if the supervised term were completely neglected?
5. In Line 362, it claims that the transformation should stay the same for CL and supervised learning when $CHN$ is used. Is this also the case for other transformation distribution choices?

**Ethical Concerns:**

["NO or VERY MINOR ethics concerns only"]

**Final Justification:**

The reviewer thinks the contrastive learning approach applied in machine unlearning is well justified in the paper, and certain theoretical results are provided. The proposed method seems to be very straightforward, and easy to follow, which may inspire future research in the area. The experimentation side may need more ablation studies to understand how each component contributes to the overall performance. Although the rebuttal response addresses partial concerns, full results should be provided in the later revisions. Therefore, the reviewer thinks the paper should be rated as "4: borderline accept".

**Limitations:**

There is no obvious limitation or negative societal impact.

**Quality:**

2

**Strengths And Weaknesses:**

Strength
1. Although contrastive learning is a known technique, it is relatively new to apply this technique in machine unlearning tasks, which relieves the constraints for forget data access, and achieve good model utility. In addition, the reviewer empirically agrees that the forget data representation is not necessarily pushed away from its original cluster, especially in a random erasing setting. The proposed approach may inspire future research in this area for better privacy and security in ML space, given the richness in contrastive learning.
2. The empirical experimental study provides a clear view of the performance equivalence compared with retrain model with multiple state-of-arts machine unlearning methods.
3. The theoretical analysis presents a lower misclassification upper bound for retain data, implying higher misclassification rate over forget data using contrastive learning.

Weakness
1. The proposed approach is a vanilla contrastive learning method, which can also be considered as model weight perturbation categories in general. There is no specific mechanism to prevent excessive or insufficient perturbation.
2. More ablation studies can be conducted to understand the behavior of the proposed approach and hyperparameter choices. Please check the question section for more details.

---

> ### Author Rebuttal · Authors · 2025-07-30
>
> We thank the reviewer for their constructive comments and for highlighting that our idea is **relatively new**, **eliminates the constraint for forget data access**, **achieves good model utility**, and **inspires future research** in privacy and security within ML. Below, we address their specific comments and questions.
>
> $
> \textcolor{white}{f^{*}}
> $
>
> **Authors' response to Q1 & W1:** We thank the reviewer for their suggestion. Our CoUn's objective function, CE + λ·CL, is designed such that the cross-entropy (CE) term preserves retain representations within their clusters, while the contrastive learning (CL) component pushes forget representations toward semantically similar retain clusters. Extending training allows the model to further refine these adjustments, reducing the average gap and improving unlearning performance. In contrast, shorter training may result in insufficient refinement, leading to suboptimal unlearning. In both cases the clustering of the retain representations will be preserved due to CE.
>
> In response to the reviewer’s suggestion, we provide experimental results in the Table below, which show that reducing training time increases the average gap, while longer training improves it.
>
> Additionally, we would like to clarify that the terms "excessive" and "insufficient" perturbation were used in reference to related works that directly modify model weights, which can lead to excessive or insufficient **weight perturbation** if not carefully controlled. For example, NoT [8] applies layer-wise weight negation. In contrast, CoUn does not directly perturb weights; instead, the CL component operates on the data representations. In our revision, we will explicitly clarify this distinction by referring to perturbation specifically in the context of **weight perturbation**.  Lastly, in Figures 7 and 8 of the paper, we demonstrate that tuning the hyperparameters will enhance the effectiveness of our proposed CoUn, thus controlling the indirect weight perturbation via learning the new forget representation adjustments toward semantically similar retain samples.
>
> **Table: Average gap for CIFAR-10 and CIFAR-100 using ResNet-18 at different training epochs.**
> | Epoch | CIFAR-10 | CIFAR-100 |
> | -------- | -------- | -------- |
> | 25    | 0.54     | 3.56     |
> | 50    | 0.25     | 1.39     |
> | 75    | 0.21     | 1.30     |
>
> $
> \textcolor{white}{f^{*}}
> $
>
> **Authors' response to Q2 & W2:** We thank the reviewer for pointing this out. In Table 2 of our paper, we provide results with tuned $\lambda$ for each experiment. Similarly, for all other baselines we tuned their hyperparameters to get the best performance. This is consistent with widely adopted practice in the literature where they report the outcome of an experiment after fine-tuning its hyperparameters.
>
> Following the reviewer’s suggestion, we present results using a fixed λ=0.5 in the Table below. Although performance decreases with a fixed λ, careful tuning is standard practice. Recommended tuning ranges are provided in Appendix C.
>
> **Table: Performance comparison of CoUn when $\lambda$ is tuned per experiment vs $\lambda$ is fixed for all experiments.**
> | Epoch | Tuned $\lambda$| Fixed $\lambda$ |
> | - | - | - |
> | CIFAR-10, ResNet-18 | 0.25 | 1.48|
> | CIFAR-100, ResNet-18 | 1.39| 1.70 |
> | TimyImageNet, ResNet-18 | 1.95 | 2.16 |
>
> $
> \textcolor{white}{f^{*}}
> $
>
> **Authors' response to Q3 & W2:** In our reported computational cost, we followed the baseline works that have been published in top-tier conferences [8,9,10,11], focusing solely on training costs under final tuned hyperparameters. Consistent with standard practice in machine unlearning literature, we did not include hyperparameter tuning costs for either CoUn or baseline methods, ensuring fair comparisons. Our goal was to provide a fair comparison of the training cost under the best-known or officially recommended hyperparameter settings, as commonly done in the baselines.
>
> $
> \textcolor{white}{f^{*}}
> $
>
> **Authors' response to Q4 & W2:** Thank you for raising this point. When training without supervision, retain samples fail to preserve cluster structure, leading to cluster overlap and notable performance degradation. Furthermore, if the classifier layer is not trained alongside the feature extractor, then relying solely on contrastive learning (CL) being applied to the feature extractor, the classifier will become misaligned with the learned features. This dramatically reduces overall performance. For instance, an experiment (CIFAR-100, ResNet-18, 10% forget ratio) without supervision resulted in an average gap of 70.56, compared to 1.39 when using CE+λ·CL.
>
> $
> \textcolor{white}{f^{*}}
> $
>
> **Authors' response to Q5 & W2:** We thank the reviewer for this observation. In the unlearning literature [8,9,10], the transformation used for training the Original and Retrain models is $CHN$, and the same transformation is also applied to the baselines. Following this convention, we used $CHN$ for training the Original, Retrain, and baseline models. Since our proposed approach incorporates contrastive learning (CL), we explored the possibility of adjusting the transformation distribution specifically for the CL component. The results presented in Figure 9 are based on the experimental setup where both the Original and Retrain models were trained using the $CHN$ transformation. Therefore, the statement in Line 362 holds only under that specific setup. In our paper revision, we will explicitly clarify that this finding is specific to that experimental configuration.

---

> > ### Comment · Reviewer_1SXk · 2025-08-01
> >
> > Thanks for the responses. Here are some reflections on the responses.
> >
> > **Regarding Q1 & W1**: The average gap at different epochs seems promising. The reviewer would encourage authors to include a figure showing how the average gap evolves at different epochs for each methods to check how efficient the proposed method may perform unlearning and converge.
> >
> > **Regarding Q3 & W2**: Although it may be a common practice in some existing literature, hyperparameter tuning itself may contribute a lot to the overall computation resource consumption. It would be good to perform more analysis on how the computation cost varies for each method regarding hyperparameter tuning.

---

> > > ### Author Response · Authors · 2025-08-01
> > >
> > > We sincerely thank the reviewer for their thoughtful reflections and encouraging feedback.
> > >
> > > $
> > > \textcolor{white}{f^{*}}
> > > $
> > >
> > > **Regarding reviewers' response to Q1 & W1:** We are pleased that the reviewer found the average gap results promising. We fully agree that visualizing the evolution of the average gap across epochs can provide valuable insight into the efficiency and convergence behavior of unlearning methods. Since we cannot include figures in the rebuttal, we will include this figure in the revised manuscript.
> > >
> > > $
> > > \textcolor{white}{f^{*}}
> > > $
> > >
> > > **Regarding reviewers' response to Q3 & W2:** We appreciate the reviewer’s follow-up on the computational cost of hyperparameter tuning. In Appendix C.1, we provided the number of hyperparameters and the tuning ranges for each across all methods, including ours. Specifically, CoUn requires tuning two hyperparameters ($\lambda$ and $\tau$)  which matches the number of hyperparameters of the competitive baseline $\ell_1$-sparse (i.e., regularization parameter, and the number of epochs it should be applied). However, we would like to highlight that very small values in the hyperparameter range (e.g., 0.0001 in $\ell_1$-sparse, as noted in Line 617) suggest high sensitivity to slight changes, thereby requiring a fine-grained search for effective tuning. This, in turn, increases the overall computational cost of hyperparameter tuning. In response to the reviewer’s suggestion, we will expand the discussion and conduct additional experiments to compare the hyperparameter tuning sensitivity and computational overhead across methods.
> > >
> > > $
> > > \textcolor{white}{f^{*}}
> > > $
> > >
> > > We are grateful for the reviewer’s constructive suggestions and recognition of our method’s potential.

---

> > > > ### Author Response · Authors · 2025-08-06
> > > >
> > > > We thank the reviewer again for the thoughtful feedback and reflections. As the rebuttal period comes to a close, we would appreciate it if the reviewer could let us know whether we have addressed their concerns. If there are any remaining questions or concerns, we would be happy to address them.

---

> > > > > ### Comment · Reviewer_1SXk · 2025-08-06
> > > > >
> > > > > Thanks for all the responses. The rebuttal response addressed most of my concerns. I would like to keep my original score to reflect the current status of the submission, and encourage authors to include their more comprehensive ablation studies in future revisions.

---

> > > > > > ### Author Response · Authors · 2025-08-06
> > > > > >
> > > > > > We thank the reviewer once again for the valuable feedback and constructive suggestions. As suggested by the reviewer, we conducted the following ablation studies during the rebuttal period and will include all of them in the revised version of our paper:
> > > > > >
> > > > > >
> > > > > >
> > > > > > * Impact of the number of training epochs on CoUn's performance
> > > > > >
> > > > > > * Sensitivity analysis of $\lambda$ across different experiments
> > > > > >
> > > > > > * Effect of removing the CE component from the objective function
> > > > > >
> > > > > >
> > > > > >
> > > > > > Regarding the experiment on the computational cost of hyperparameter tuning, we are unable to complete it within the few remaining days of the rebuttal period due to time constraints. However, we have provided our intuition that this cost depends on the number of hyperparameters and the algorithm's sensitivity to them. Based on this, we infer that the cost would be lower for CoUn compared to $\ell_1$-sparse as a competitive baseline. In line with the reviewer's suggestion, all these results will be reflected in the revised version.
> > > > > >
> > > > > >
> > > > > >
> > > > > > We are grateful that we were able to address most of the reviewer’s concerns. Please let us know if there are any specific remaining points we can clarify within the rebuttal period.

---

### Official Review · Reviewer_CMM6 · 2025-07-01

**Clarity:** 2
**Significance:** 2
**Originality:** 2
**Rating:** 4
**Confidence:** 5

**Summary:**

This paper proposes to perform both contrastive learning and supervised learning on retain data to improve the unlearning effects.

**Questions:**

See the weaknesses.

**Ethical Concerns:**

["NO or VERY MINOR ethics concerns only"]

**Final Justification:**

After the authors' rebuttal, I raised my rating from 3 to 4.

**Limitations:**

yes

**Paper Formatting Concerns:**

no paper formatting concerns

**Quality:**

3

**Strengths And Weaknesses:**

Strengths:

1. The proposed method is simple yet effective. It incorporates contrastive loss in the finetuning loss and presents clear performance improvement compared to standard fine-tuning.

2. Experiments are comprehensive.

3. The proposed method surpasses baseline methods in multiple unlearning scenarios.

4. Notations, figures, tables, and theorems are very clear. The paper is well-organized.



Weaknesses:

1. Sec 3.2 emphasizes that conducting CL on retain data indirectly pushes forget representations toward high-similarity retain clusters, and this can improve forget quality. Here, more empirical results are needed to support this motivation and assumption. Specifically, for a model A trained on the whole dataset and a model B retrained on only retain data, what are the differences in the similarity between forget representations and the "high-similarity" retain clusters? For comparison, there should be two more charts of the Model A before unlearning in Figure 1. Moreover, a rigid table of statistical values is more desired than Figures 1 and 3 for a valid presentation.

2. Furthermore, it's unclear why pushing forget representations toward other retain samples can improve forget quality, which is defined as forget accuracy or MIA efficacy. Since approximate unlearning sets retraining as a standard, improving forget quality should refer to a closer gap in forget accuracy or MIA efficacy. Regarding this, will the action of pushing forget representations increase or decrease these two metrics? Is this action imitating Retraining? Otherwise,  how do you determine whether there will be an improvement?

3. The theoretical analysis says that CoUn yields a higher misclassification rate on forget data compared to retain data. This property is not surprising since the CoUn performs both contrastive learning and supervised learning on retain data, and the forget data are treated as test data. However, this analysis does not support the reason why CoUn provides better empirical unlearning performance, since from Table 2 all other baseline methods have higher retain accuracy and lower forget accuracy. The authentic significance of such theoretical analysis is quite questionable.

4. Although CoUn consistently shows the best Avg. Gap in the figures and tables, different settings yield different second-best methods. For example, recent methods like Salun and NoT sometimes outperform NegGrad, while in other cases they do not. Yet, each new method claims to outperform previous ones when introduced. This inconsistency in performance raises concerns about the fairness and accuracy of reproducing baseline methods.

5. In some cases CoUn's superiority in Avg. The gap is marginal, but it costs more computation resources.

---

> ### Author Rebuttal · Authors · 2025-07-30
>
> We thank the reviewer for their valuable feedback. We are pleased they find our proposed method **simple yet effective**, presenting a **clear performance improvement**, and that the **experiments are comprehensive**. We also appreciate the reviewer's acknowledgment of our **clear notations, figures, tables, and theorems**, and that the **paper is well-organized**. Below, we address their specific comments and questions.
>
> $
> \textcolor{white}{f^{*}}
> $
>
> **Authors' response to W1:** We thank the reviewer for their suggestion. Following the reviewer’s suggestion, we have generated the t-SNE visualization for the Original model trained on the complete dataset (retain + forget data). This Original model serves as a starting checkpoint for both class-wise and random data forgetting scenarios; thus, its the same for both scenarios. We will add the t-SNE visualization of the Original model in the paper. Visually, this plot resembles Figures 1 or 3 (left) but with 10 clusters and no misclassifications. From another perspective, it also resembles the random forgetting scenario plots (right), again without misclassifications.
>
> In response to the reviewer’s suggestion to provide a more rigid statistical comparison between forget representations and retain clusters, we grouped the forget samples by their true class labels and, for each group, computed the average Euclidean distance (L2) from its samples to all retain class centroids. This yielded a per-class distance profile showing how far forget representations lie from each retain cluster. To enable comparison across different models, we then applied normalization on each group’s averaged distances. The Table below summarizes the results for CIFAR-10, ResNet-18, and the statistics for 'truck' forget samples in a 10% random forgetting (same setup as Table 1 in our paper). The findings show that forget representations in CoUn are consistently closer to semantically similar retain clusters, and more importantly, CoUn achieves distance statistics that are closer to those of the Retrain model compared to other baselines. The smaller the distance means higher semantic similarity. From the Table below, we can see that 'truck' samples have the highest semantic similarity with 'automobile'. These results will be reflected in the paper.
>
> **Table: L2 distances of forget 'truck' to retain centroids. The most semantically similar clusters to 'truck' samples are presented. Experiments conducted using CIFAR-10 and ResNet-18. The difference ($\Delta$) and the (best) average difference between each method and Retrain are reported.**
> |Forgetting Scenario|Method|Automobile|Airplane|Ship|Avg. Diff. ↓|
> |-|-|-|-|-|-|
> |Random (10%)|Original|0.93|0.97|0.96|-|
> ||Retrain|0.90 (0.00)|0.96 (0.00)| 0.95 (0.00)|0.000|
> ||FT|0.86 (0.04)|0.94 (0.02)|0.91 (0.04)|0.033|
> ||**CoUn (Ours)**|0.87 (0.03)|0.96 (0.00)|0.93 (0.02)|**0.017**|
>
> Furthermore, we kindly refer the reviewer to Table 1 in our paper, which presents part of the confusion matrix for forget samples under both forgetting scenarios, comparing predictions by the Retrain model (oracle) and our CoUn method. We can see that the predictions are based on semantic similarity. Smaller differences between their prediction distributions indicate better forgetting quality by CoUn. For example, Table 1 shows the top-4 predictions for forget samples labeled 'truck'. While Table 1 does not include the Original model or other baselines, we have conducted those evaluations and will include them in the paper. Below we show the top-4 predictions for the forget 'truck' samples. The Original model classifies all 'truck' forget samples correctly (100%) as expected, as it was trained on them. In contrast, Retrain and other methods (FT, NegGrad+, $\ell_1$-sparse, SalUn, NoT, CoUn), under random forgetting, predict most forget samples as 'truck', 'automobile', 'airplane', and 'ship', based on semantic similarity with retain clusters. CoUn consistently shows the smallest deviation from Retrain's predictions, highlighting its superior imitation of the Retrain model. To clarity, if a 'truck' forget sample is predicted as another label, then it means its representation was pushed toward a semantically similar retain cluster that is different than the forget's sample cluster; yet, if it is still predicted as 'truck', then it means the forget representation was pushed to a retain representation within the same cluster.
>
> **Table: Predictions of forget 'truck' samples based on most semantically similar classes. The difference ($\Delta$) and the (best) average difference between each method and Retrain are reported.**
> |Forgetting Scenario|Method|Truck|Automobile|Airplane|Ship|Avg. Diff. ↓|
> |-|-|-|-|-|-|-|
> |Random (10%), CIFAR-10, ResNet-18|Original|100.00|0.00|0.00|0.00|-|
> ||Retrain|97.42 (0.00)|1.23 (0.00)|0.38 (0.00)|0.40 (0.00)|0.00|
> ||FT|98.12 (0.70)|0.75 (0.48)|0.36 (0.02)|0.32 (0.08)|0.32|
> ||NegGrad+|97.86 (0.44)|0.97 (0.26)|0.42 (0.04)|0.30 (0.10)|0.21|
> ||$\ell_1$-sparse|98.06 (0.64)|0.85 (0.38)|0.30 (0.08)| 0.38 (0.02)|0.28|
> ||SaUn|96.76 (0.66)|0.88 (0.35)| 0.34 (0.04)|0.34 (0.06)| 0.28|
> ||NoT|97.88 (0.56)|0.89 (0.34)| 0.40 (0.02)| 0.32 (0.08)| 0.25|
> ||**CoUn (Ours)**|97.84 (0.42)| 0.99 (0.24)| 0.32 (0.06)|0.42 (0.02)|**0.19**|
>
> $
> \textcolor{white}{f^{*}}
> $
>
> **Authors' response to W2:** We thank the reviewer for raising this point. Forget quality is typically assessed using forget accuracy: the drop in accuracy on forget data, and MIA (Membership Inference Attack) efficacy: how indistinguishable forget samples become from non-training samples.
>
> In Figure 1 and Table 1, we showed that the Retrain model classifies forget samples into clusters of retain samples that exhibit the highest semantic similarity to them. Now, since in CoUn we push forget representations towards other retain representations that exhibit highest semantic similarity to them, this results in an unlearned model that imitates the behavior of Retrain model in classifying forget samples.
>
> When forget samples are pushed into semantically similar retain clusters, this reduces their distinctiveness, making them behave more like non-training sample (improving MIA). At the same time, pushing forget samples away from their original clusters increases misclassification rates, thereby decreasing forget accuracy (or equivalently, increasing unlearn accuracy, defined as 1 - forget accuracy).
>
> Therefore, improving forget quality involves increasing MIA efficacy and decreasing forget accuracy. Our CoUn method achieves this by imitating the Retrain model's semantic-based clustering of forget data, outperforming other methods without compromising utility.
>
> $
> \textcolor{white}{f^{*}}
> $
>
> **Authors' response to W3:** We thank the reviewer for the opportunity to clarify this point.
>
> 1. Our theoretical analysis aims to formally characterize this intuitive behavior and offer a provable foundation for reasoning about misclassification on forget data. Specifically, our analysis quantifies how feature representations learned from retain data fail to generalize to forget data. This helps ensure that forgetting is not only effective empirically but also theoretically justified.
>
> 2. We believe that theoretical analysis of the misclassification rate contributes to the overall assessment of unlearning performance. In particular, as mentioned in Lines 231–234, achieving a higher misclassification rate on forget data while maintaining a low misclassification rate on retain data supports both good forget quality and high model utility. A low misclassification rate on retain data indicates high retain accuracy, while a higher misclassification rate on forget data corresponds to lower forget accuracy and thus improved unlearn accuracy.
>
> 3. In fact, Table 2 **does not** indicate that "all other baseline methods have higher retain accuracy and lower forget accuracy" compared to CoUn. On the contrary, CoUn consistently achieves retain accuracy and unlearn accuracy (i.e., 1 - forget accuracy) closer to the Retrain model, which represents the ideal target performance.
>
> $
> \textcolor{white}{f^{*}}
> $
>
> **Authors' response to W4:** We acknowledge this variability, which aligns with results reported in [11] (NeurIPS 2024). For example, rankings among NegGrad+, $\ell_1$-sparse, and SalUn frequently shift across evaluations.  In [11, Table 12] using CIFAR-10 and ResNet-18, the ranking based on average gap of methods differs across evaluations as follows:
>
> * [11, Table 12a]: NegGrad+ < $\ell_1$-sparse < SalUn.
> * [11, Table 12b]: $\ell_1$-sparse < NegGrad+ < SalUn.
> * [11, Table 12c]: SalUn < NegGrad+ < $\ell_1$-sparse.
>
> Similarly, [11, Table 10] shows:
> * [11, Table 10a]: NegGrad+ < SalUn < $\ell_1$-sparse.
> * [11, Table 10b]: NegGrad+ < SalUn < $\ell_1$-sparse.
> * [11, Table 10c]: $\ell_1$-sparse < SalUn < NegGrad+.
>
> We carefully tuned the hyperparameters for all baseline methods, including CoUn, to ensure they yield the best performance. Further, we compared all methods under equal computational budgets to ensure experimental fairness (see Figure 5).
>
> $
> \textcolor{white}{f^{*}}
> $
>
> **Authors' response to W5:** We kindly refer the reviewer to Figure 5 in our paper, where we demonstrated that CoUn consistently outperforms baseline methods under equal computational budgets. For clarity, we also include the Table below, which presents the exact percentage improvements from Figure 5 (at 50% forget ratio) under matched computational cost. This highlights that the contrastive learning component substantially improves performance without additional computational cost.
>
> **Table: Percentage improvement in Avg. Gap of CoUn and baselines relative to FT, under matched computational cost.**
> |Method| Avg. Gap($\downarrow$) | Improvement ($\uparrow$) | Comp. cost (PFLOPs) ($\downarrow$)|
> |-|-|-|-|
> |FT|6.89|-| 4.19|
> |NegGrad+ |3.70|+46.30%|5.58|
> |$\ell_1$-sparse|2.11|+69.38%|4.19|
> |NoT|5.66| +17.85%|4.19|
> |**CoUn (Ours)**|**1.87**|**+72.86%**|**4.13**|

---

> > ### Comment · Reviewer_CMM6 · 2025-08-04
> >
> > I appreciate the authors’ response. Your clarifications have addressed most of my concerns. My remaining concern is about this field: as I mentioned in W4, evaluation on classification tasks has become a game of fitting five metrics, and the superiority of each new method does not seem to be entirely valid.
> >
> > I believe that under the current rules, this paper does not appear to have any significant flaws. Therefore, I will raise my score from 3 to 4, which I believe is fair to the authors.
> >
> > However, as a reviewer, it is difficult for me to determine how important the improvement in the average gap truly is for the field of unlearning. I will continue to consider the opinions of other reviewers and the AC.

---

> > > ### Author Response · Authors · 2025-08-04
> > >
> > > We sincerely thank the reviewer for their updated evaluation and for acknowledging that our clarifications addressed most of the concerns. We also appreciate the candid reflection on the current state of evaluation practices in the unlearning field.
> > >
> > > While the metrics we used follow standard practices in the literature, we agree with the reviewer that developing new evaluation for classification could be a valuable direction for future work. Further, in our experiments we did our best to provide a fair comparison with the baselines.
> > >
> > > Once again, we are grateful for your engagement with our work and truly appreciate your thoughtful re-evaluation.

---

### Official Review · Reviewer_z7vM · 2025-07-02

**Clarity:** 3
**Significance:** 2
**Originality:** 2
**Rating:** 4
**Confidence:** 3

**Summary:**

This paper tackles the problem of machine unlearning by incorporating contrastive learning. Specifically, the proposed method applies contrastive learning to adjust forget representations and supervised learning to maintain retain representations within their clusters sharing similar semantics. Experimental results show the effectiveness of the proposed method across small image datasets with CNNs and a ViT.

**Questions:**

Please address concerns in Weaknesses above.

**Ethical Concerns:**

["NO or VERY MINOR ethics concerns only"]

**Final Justification:**

The authors successfully addressed my concerns and promised to release their code for reproducibility. Hence, I raise my score accordingly.

**Limitations:**

yes

**Paper Formatting Concerns:**

nothing special

**Quality:**

3

**Strengths And Weaknesses:**

**Strengths**

+ Machine unlearning is an important topic for deployment of machine learning models, and this work aims to make a step forward in this field.

+ The idea is simple and supported with some theoretical results.


**Weaknesses**

- I agree with the importance of machine unlearning, but skeptical whether the current evaluation strategy is proper to assess machine unlearning methods. it is possible that the information about "training data to forget" still remain in the new model but its accuracy matches with the gold-standard retrain model by another reason, e.g., losing other information. In other words, accuracy matching does not imply that two models exhibit exactly the same knowledge, and it might not satisfy GDPR; rather, prediction matching would be more suitable.

- Basically, I don't think classification is a proper task to assess forgetting of specific data; rather, reconstruction for specific data would be more suitable. For example, it is possible that forgetting some data allows the new model to have more capacity to better keep the information of other training data, resulting in better classification accuracy. In this case, unlearning would increase classification accuracy.

- Contrastive learning has been applied to many fields in machine learning, so the usage of contrastive learning without much adaptation as done in this paper has a limited novelty.

- Theorem 1 is originated from Huang et al. [34], which considers pure contrastive learning settings; however, the proposed method learns through both contrastive learning and supervised learning, so Theorem 1 might not hold.

- No forgetting measure in experiments, which has commonly been employed in prior works.

- Performance gain seems to be consistent but not so significant, raising concerns on the trade-off between the additional cost introduced by contrastive learning versus the gain.

- Experiments are done in small image datasets, so its scalability and generalizability to other datasets/domains are not clear.

- Experimental results might not be reproducible as the description on experimental settings is not sufficient and the provided code is only for a portion of experiments. For example, I couldn't find the configuration for ViT experiments, and "ViT" is not enough to figure out and replicate its specification.

- A similar work by [Lee et al.] is not cited/discussed.

[Lee et al.] Contrastive Unlearning: A Contrastive Approach to Machine Unlearning. arXiv 2024.

- typo: NIPS -> NeurIPS

---

> ### Author Rebuttal · Authors · 2025-07-30
>
> We thank the reviewer for their constructive feedback. We are pleased the reviewer finds our paper **simple**, **supported by theoretical analysis**, **demonstrating effectiveness through experimental results**, and acknowledges that **our work advances an important topic**. Below, we address their specific comments and questions.
>
> $
> \textcolor{white}{f^{*}}
> $
>
> **Authors' response to W1:** We thank the reviewer for the opportunity to clarify this point.
> 1. Indeed, evaluating machine unlearning methods requires metrics **beyond accuracy alone**, as accuracy matching alone does not guarantee genuine forgetting or GDPR compliance. We would like to emphasize that our evaluation follows standard practices established in recent machine unlearning literature (e.g., [8] @ CVPR 2025, [9] @ ICLR 2024, [11] @ NeurIPS 2024, [10] @ NeurIPS 2023). Consistent with these studies, we report **Membership Inference Attack (MIA) alongside retain accuracy, unlearn accuracy, and test accuracy**. The MIA success rate specifically measures how effectively forget data samples are identified as non-training in the unlearned model. Note: unlearn accuracy = 1 - forget accuracy. Importantly, our dataset is explicitly partitioned into retain and forget subsets. Since, we evaluate both retain and forget accuracies then accuracy matching can not happen due to unintended loss of retain data (i.e., "losing other information"), as this would lead to a drop in retain accuracy. However, our results show that retain accuracy is maintained, while forget accuracy drops, indicating that the loss of information is localized to the forget data. This pattern is further confirmed by the MIA scores, which consistently show reduced influence of the forget data post-unlearning.
>
> 2. We also concur that **prediction-level comparison** offers a finer-grained assessment of model behavior. Accordingly, please refer to **Table 1 in our paper**, where we incorporated prediction-level comparison metrics to evaluate alignment with the gold-standard Retrain and FT baseline. Additionally, we provide extended results in the Table below showing prediction-level comparisons for a single forget class across additional baselines and datasets. These results consistently demonstrate that CoUn achieves the lowest average prediction divergence compared to the gold-standard Retrain model. This supports our claim that CoUn effectively mimics the clustering behavior of the Retrain model with respect to forget data based on semantic similarity better than the baselines.
>
> **Table: Predictions of forget 'truck' samples based on most semantically similar classes. The difference ($\Delta$) and the (best) average difference between each method and Retrain are reported.**
> |Forgetting Scenario|Method|Truck|Automobile|Airplane|Ship|Avg. Diff. ↓|
> |-|-|-|-|-|-|-|
> |Random (10%), CIFAR-10, ResNet-18|Original|100.00|0.00|0.00|0.00|-|
> ||Retrain|97.42 (0.00)|1.23 (0.00)|0.38 (0.00)|0.40 (0.00)|0.00|
> ||FT|98.12 (0.70)|0.75 (0.48)|0.36 (0.02)|0.32 (0.08)|0.32|
> ||NegGrad+|97.86 (0.44)|0.97 (0.26)|0.42 (0.04)|0.30 (0.10)|0.21|
> ||$\ell_1$-sparse|98.06 (0.64)|0.85 (0.38)|0.30 (0.08)| 0.38 (0.02)|0.28|
> ||SaUn|96.76 (0.66)|0.88 (0.35)| 0.34 (0.04)|0.34 (0.06)| 0.28|
> ||NoT|97.88 (0.56)|0.89 (0.34)| 0.40 (0.02)| 0.32 (0.08)| 0.25|
> ||**CoUn (Ours)**|97.84 (0.42)| 0.99 (0.24)| 0.32 (0.06)|0.42 (0.02)|**0.19**|
> ||
> |Random (10%), CIFAR-100, VGG-16|Original| 100.00|0.00|0.00|0.00|-|
> ||Retrain|49.53 (0.00)|15.18 (0.00)|10.06 (0.00)|3.42 (0.00)|0.00|
> ||FT|60.53 (11.00)|9.11 (6.07)|9.68 (0.38) |3.61 (0.19)|4.41|
> ||NegGrad+|41.94 (7.59)|13.47 (1.71)|12.71 (2.65)|4.74 (1.32)|3.32|
> ||$\ell_1$-sparse|57.69 (8.16)|13.47 (1.71)|10.06 (0.00)|2.85 (0.57)|2.61|
> ||SaUn|55.16 (5.63)|13.52 (1.66)| 9.68 (0.38)|2.63 (0.79)|2.12|
> ||NoT|51.23 (1.70)|14.61 (0.57)|9.68 (0.38)|3.61 (0.19)|0.71|
> ||**CoUn (Ours)**|50.15 (0.62)|14.67 (0.51)|10.09 (0.03)|3.28 (0.14)|**0.33**|
>
> $
> \textcolor{white}{f^{*}}
> $
>
> **Authors' response to W2:** We thank the reviewer for suggesting this potential area for exploration. While we agree that reconstruction-based evaluations may provide additional insights into data-specific forgetting, classification remains widely adopted in the top-tier machine unlearning literature [8,9,10,11].
>
> Importantly, proper unlearning should not result in improved generalization (i.e., higher classification test accuracy). Although not explicitly reported in the main results, we confirm that the test accuracy of the Original model is higher than that of the Retrain model, as expected. This drop reflects the reduced training set used by the Retrain model (i.e., retain data only), which naturally leads to a decrease in accuracy. Moreover, even if removing some data allows a model to better utilize its capacity and improve performance on the remaining data, this benefit would apply equally to both the Retrain and the unlearned models—since both are trained without access to the forget data. Also, in unlearning evaluations, we focus on the relative difference between the unlearned model and the Retrain model, rather than absolute improvements in classification accuracy.
>
> Introducing reconstruction-based metrics targeting specific samples could offer a complementary view of forgetting and is an interesting direction. Due to time constraint, we will explore and discuss this direction in future work.
>
> $
> \textcolor{white}{f^{*}}
> $
>
> **Authors' response to W3:** Even though contrastive learning (CL) is widely adopted, its application to machine unlearning remains novel. Our contribution lies in **strategically integrating CL with supervised learning**, **utilizing only retain data** to guide the unlearned model toward the behavior of the Retrain model **through the lens of semantic similarity**. Furthermore, **our modular CL design allows easy incorporation into existing and future unlearning methods**, enhancing alignment fidelity broadly.
>
> $
> \textcolor{white}{f^{*}}
> $
>
> **Authors' response to W4:** We thank the reviewer for raising this point. It is true that Theorem 1 is derived under the assumption that the feature extractor is trained using contrastive learning, and the classifier head is fine-tuned on a downstream task. However, when the feature extractor is trained using a combination of contrastive and supervised learning, the misclassification rate bound provided in Theorem 1 can still serve as a valid upper bound. Therefore, the inequality stated in Line 258 continues to hold, and our justification in Lines 257–262 remains applicable.
>
> $
> \textcolor{white}{f^{*}}
> $
>
> **Authors' response to W5:** We evaluate forgetting using two widely adopted metrics: unlearn accuracy: indicating model performance on forget data, and MIA: reflecting how effectively forget data is excluded from training. These metrics align with what recent top-tier unlearning literature [8,9,10,11] have used in their empirical evaluation. Detailed explanations of these metrics appear in Lines 48–52 and the Evaluation Metrics section.
>
> $
> \textcolor{white}{f^{*}}
> $
>
> **Authors' response to W6:** We would like to refer the reviewer to Figure 5 in our paper, demonstrating that under equal computational budgets, our CoUn method consistently and significantly outperforms baseline methods. The Table below provides exact improvements from Figure 5 (50% forget ratio) under matched computational costs, emphasizing CoUn's superior performance without additional computational expense.
>
> **Table: Improvement in average gap of CoUn and baselines relative to FT, under matched computational cost.**
> |Method|Avg. Gap ($\downarrow$)|Improvement ($\uparrow$)|Comp. cost (PFLOPs) ($\downarrow$)|
> |-|-|-|-|
> |FT|6.89|-|4.19|
> |NegGrad+|3.70|+46.30%|5.58|
> |$\ell_1$-sparse |2.11|+69.38%|4.19|
> |NoT|5.66|+17.85%|4.19|
> |**CoUn (Ours)**|**1.87**|**+72.86%**|**4.13**|
>
> $
> \textcolor{white}{f^{*}}
> $
>
> **Authors' response to W7:** We acknowledged this limitation explicitly in Appendix G (Limitations) of our paper. We stated that "our evaluation is limited to relatively small datasets (CIFAR-10/100, TinyImageNet). Future work could explore larger datasets like ImageNet." Additionally, [8] @ CVPR 2025 mentioned in their Limitations section that they evaluated on small datasets.
>
> Moreover, our current experimental setup on CIFAR-10/100 and TinyImageNet aligns with [11] @ NeurIPS 2024, which used the same datasets (CIFAR-10/100 and TinyImageNet) for their experiments. Lastly, these datasets align to some degree with [8,10], which also evaluated on CIFAR-10/100. Thus, our dataset selection ensures that our evaluation setup is consistent with established benchmarks, enabling meaningful comparisons with state-of-the-art methods.
>
> $
> \textcolor{white}{f^{*}}
> $
>
> **Authors' response to W8:** We thank the reviewer for identifying this. We would like to note that the complete ViT model configuration is provided in our supplementary code (utils/model.py, Line 15):  "patch_size=(4, 4), num_classes=num_classes, dim=512, mlp_dim=1024, dim_head=64, depth=6, heads=12, dropout=0.1, emb_dropout=0.1".
>
> We acknowledge this omission in the paper and we will update it accordingly. Full training details appear in Lines 283-288 and in Appendix C. ViT-specific configurations for baselines, such as NoT, are mentioned explicitly (see Line 621). The ViT architecture file is included in our supplementary code, and we also cited the ViT paper (see Line 266).
>
> $
> \textcolor{white}{f^{*}}
> $
>
> **Authors' response to W9:** We would like to clarify that the work by [Lee et al.] is indeed cited in our paper as Reference [32]. We discussed, compared, and contrasted their approach in the Related Work section, with further detail provided in Appendix A. Moreover, we present in Table 4 a direct experimental comparison between CU [32] and our method CoUn, demonstrating that CoUn significantly outperforms CU.
>
> $
> \textcolor{white}{f^{*}}
> $
>
> **Authors' response to W10:** Thank you for highlighting this typo. We will correct it.

---

> > ### Comment · Reviewer_z7vM · 2025-08-04
> >
> > Thank you for your response. While most of my major concerns have been addressed, I still have a concern regarding reproducibility. I wonder if the authors have any plans to release their code.

---

> > > ### Author Response · Authors · 2025-08-04
> > >
> > > We sincerely thank the reviewer for taking the time to review our responses and for their constructive feedback.
> > >
> > > Yes, we do plan to release the code. In addition, we have already included the code in the supplementary materials, which will become publicly available upon the public release of the rebuttal.
> > >
> > > We hope this addresses your concern regarding reproducibility. We would appreciate it greatly if you can re-evaluate our paper and check whether we have addressed your comments in a satisfactory manner.

---

> > > > ### Author Response · Authors · 2025-08-06
> > > >
> > > > We want to thank the reviewer for taking the time to review our replies to their comments and questions. As the rebuttal period comes to a close, we would appreciate it if the reviewer could let us know whether we have addressed their concerns. If there are any remaining questions or points, we would be happy to address them.

---

> > > > ### Comment · Reviewer_z7vM · 2025-08-06
> > > >
> > > > The current instructions in the code are not sufficiently detailed to replicate experimental results in the paper. The authors are strongly encouraged to provide clearer and more comprehensive instructions for reproducibility. Assuming that this work is replicable with the code to be released, I will raise my score.

---

> > > > > ### Author Response · Authors · 2025-08-06
> > > > >
> > > > > We thank the reviewer for the constructive feedback. In line with the suggestion, we will enhance the codebase with clearer and more detailed instructions to facilitate full reproducibility of our work. We are committed to making our experiments easy to replicate and greatly appreciate the reviewer’s willingness to re-evaluate our work.

---

### Decision · Program_Chairs · 2025-09-17

**Decision:**

Accept (poster)

**Comment:**

**Summary:** This paper introduces CoUn, a machine unlearning framework that leverages contrastive learning in combination with supervised learning on retained data. The central idea is to indirectly adjust forget representations based on semantic similarity to retain samples, thereby emulating the behavior of retraining from scratch on retain-only data. The authors provide both theoretical insights and extensive experiments across multiple datasets and architectures, demonstrating that CoUn achieves performance close to the retrain baseline while surpassing existing unlearning methods. The approach is modular and can also be integrated with other baselines to enhance their unlearning effectiveness.


**Decision:** Initially, the reviewers raised several concerns, including the limited novelty of using contrastive learning, clarity of the theoretical analysis, scalability of experiments beyond small datasets, and reproducibility of results. During the rebuttal, the authors responded with substantial clarifications, additional empirical studies, and commitments to release detailed code. Most reviewers raised their scores after the rebuttal, acknowledging that the paper makes a meaningful and technically sound contribution. Given the importance of machine unlearning, the clarity of the proposed framework, and the convincing experimental evidence, I recommend acceptance.